# A novel power-amplified jumping behavior in larval beetles (Coleoptera: Laemophloeidae)

**Matthew A. Bertone** [1‡*], **Joshua C. Gibson** [2‡], **Ainsley E. Seago** [3], **Takahiro Yoshida** [4], **Adrian A. Smith** [5,6]

**1** Department of Entomology and Plant Pathology, North Carolina State University, Raleigh, North Carolina, United States of America, **2** Department of Entomology, University of Illinois at Urbana-Champaign, Urbana-Champaign, Illinois, United States of America, **3** Section of Invertebrate Zoology, Carnegie Museum of Natural History, Pittsburgh, Pennsylvania, United States of America, **4** Systematic Zoology Laboratory, Graduate School of Science, Tokyo Metropolitan University, Hachioji City, Tokyo, Japan, **5** Research & Collections, North Carolina Museum of Natural Sciences, Raleigh, NC, United States of America, **6** Biological Sciences, North Carolina State University, Raleigh, North Carolina, United States of America

◔ These authors contributed equally to this work.
‡ MAB and JCG are joint first authors on this work.
* matt_bertone@ncsu.edu

**Data Availability Statement:** All relevant data are within the manuscript and its Supporting Information files.

## Abstract

Larval insects use many methods for locomotion. Here we describe a previously unknown jumping behavior in a group of beetle larvae (Coleoptera: Laemophloeidae). We analyze and describe this behavior in *Laemophloeus biguttatus* and provide information on similar observations for another laemophloeid species, *Placonotus testaceus*. *Laemophloeus biguttatus* larvae precede jumps by arching their body while gripping the substrate with their legs over a period of 0.22 ± 0.17s. This is followed by a rapid ventral curling of the body after the larvae releases its grip that launches them into the air. Larvae reached takeoff velocities of 0.47 ± 0.15 m s-1 and traveled 11.2 ± 2.8 mm (1.98 ± 0.8 body lengths) horizontally and 7.9 ± 4.3 mm (1.5 ± 0.9 body lengths) vertically during their jumps. Conservative estimates of power output revealed that some but not all jumps can be explained by direct muscle power alone, suggesting *Laemophloeus biguttatus* may use a latch-mediated spring actuation mechanism (LaMSA) in which interaction between the larvae's legs and the substrate serves as the latch. MicroCT scans and SEM imaging of larvae did not reveal any notable modifications that would aid in jumping. Although more in-depth experiments could not be performed to test hypotheses on the function of these jumps, we posit that this behavior is used for rapid locomotion which is energetically more efficient than crawling the same distance to disperse from their ephemeral habitat. We also summarize and discuss jumping behaviors among insect larvae for additional context of this behavior in laemophloeid beetles.

## Introduction

The extraordinary evolutionary success of holometabolous insects can be partially attributed to their partitioned life history: immatures (larvae) are often soft-bodied and minimally

**Funding:** TY was partly supported by Research Fellowships of the Japan Society for the Promotion of Science for Young Scientists (JSPS Research Fellowships for Young Scientists, PD: JP19J00167). https://www.jsps.go.jp/english/e-pd/index.html The funders had no role in study design, data collection and analysis, decision to publish, or preparation of the manuscript.

**Competing interests:** The authors have declared that no competing interests exist.

mobile, adapted for feeding and growth, while adults are often highly mobile, enabling dispersal and mate seeking. This generally sedentary lifestyle exhibited by many larvae makes them highly attractive targets for predators and parasites. Holometabolous insects have evolved a number of solutions to the problem of larval self-defense (e.g. concealed habitats, chemical defense, parental care [1–3], but few active means of escaping predators. Rapid locomotion is inherently difficult during this life stage; terrestrial larvae are typically plump and slow-moving, with short legs (e.g. Lepidoptera, most Coleoptera, and many Hymenoptera) or no legs at all (Diptera, most Hymenoptera, and some Coleoptera). Exceptions include a few groups containing active predators (e.g. Carabidae and Chrysopidae) and triungulin or planidial larvae, in which the highly mobile first instar locates a host before reverting to a largely immobile, parasitic form in subsequent instars (referred to as hypermetamorphosis) [4].

Despite these limitations, several insect lineages have evolved distinctive methods of rapid larval locomotion without using legs at all. The larvae of some dune-dwelling tiger beetles (Coleoptera: Cicindelidae) use wind-propelled, wheel-like locomotion [5], derived from an ancestral tendency to flip or somersault when attacked by parasitoid wasps [6]. The jumping ability of some fly larvae has been informally recognized for centuries, e.g. cheese skipper maggots (Piophilidae: *Piophila casei* (L.)), whose vigorous activity has long marked the quality of a Sardinian cheese known as *casu marzu* [7–9]. Subsequent studies have found that larval jumping is widespread in holometabolous insects, including Lepidoptera, Hymenoptera, and Diptera (Table 1 and the references within). However, it was not until relatively recently that the actual physiological and kinematic mechanisms underlying larval jumping have begun to be adequately investigated.

Maitland [28] described "the only known example of jumping by a soft-bodied legless organism," in larvae of the Mediterranean fruit fly (Tephritidae: *Ceratitis capitata* (Wiedemann)) (We note that, Swammerdam [7] notwithstanding, the first detailed description of jumping locomotion in a fly larva seems to be Camazine's [12] study of *Mycetophila cingulum* Meigen). *Ceratitis* larvae achieve jumps of up to 150 times their body length by curling into a loop and pumping hemolymph into the abdominal segments until the resulting turgor pressure is sufficient for bodily propulsion when the loop is released [28]. This body loop is anchored by attachment of the mandibles to the sclerotized anal plate, effectively forming a latch-mediated spring [46]. All other reported cases of jumping in maggots appear to work in a similar way, albeit with varying attachment mechanisms: piophilids also form a ventral loop with mandibular-anal attachment, but mycetophilid larvae bend dorsally rather than ventrally, anchoring the thorax to the abdominal tergites with a velcro-like array of interlocking pegs [12]. Farley et al. [13] demonstrated that jumping larvae of the gall midge *Asphondylia* sp. anchor their body loop by connecting two regions of cuticle bearing velcro-like microstructures.

Certain case-bearing or enclosed Lepidoptera larvae are capable of short, rapid hops, by ventrally deflexing the body, increasing turgor pressure in selected body segments, then striking the interior of their case (Thyrididae: *Calindoea trifascialis* (Moore)) or seedpod (Tortricidae: *Cydia deshaisana* (Westwood); Pyralidae: *Emporia melanobasis* Balinsky) [31]. Hymenopteran "jumping galls" (Cynipidae: *Neuroterus saltatorius* Edwards) use a similar mechanism, although it is unclear whether the body loop is anchored or how the tightly-enclosed, conglobulated larva is able to displace enough hemolymph to store adequate energy for powering jumps [35]. All evidence from the above cited studies indicates that these are not escape jumps meant to evade predators, but rather a means of dispersal toward optimal pupation sites, e.g. away from direct sunlight.

Only a few of these larval jumping behaviors have been recorded using high-speed videography, which allows for precise descriptions of takeoff sequences and measurements of

**Table 1. Taxonomic distribution of jumping behavior among insect larvae.**

| Order | Family | Species | Life Stage | Maximum Distance | Speed | Mechanism | Host or substrate | Citation |
|---|---|---|---|---|---|---|---|---|
| Diptera | Acroceridae | *Ogcodes pallipes* Latreille | first instar/ planidium | | | substrate-anchored cercal spring | active among host habitats; endoparasite of spider | [10] |
| | | *Ogcodes rufoabdominalis* Cole | first instar/ planidium | | | substrate-anchored cercal spring | active among host habitats; endoparasite of spider | [11] |
| | | *Pterodontia* sp. | first instar/ planidium | | | substrate-anchored cercal spring | active among host habitats; endoparasite of spider | [10] |
| | Cecidomyiidae | *Asphondylia* sp. | third instar | 121 mm | 0.85 m/s | self-anchored loop, ventral | galls | [12, 13] |
| | | *Contarinia inouyei* Mani | third instar | | | various self-anchored loops | bud galls | [14] |
| | | *Contarinia tritici* Kirby | third instar | | | various self-anchored loops | bud galls | [15] |
| | | *Tricholaba trifolii* Rübsaamen | third instar | | | various self-anchored loops | inquilines in galls of *Dasineura* (Cecidomyiidae) | [16] |
| | Chloropidae | *Cadrema pallida* (de Meijere) | unknown | | | self-anchored loop, ventral* | decaying organic matter | [17] |
| | Clusiidae | Unknown | unknown | | | self-anchored loop, ventral* | saproxylic | [18] |
| | Drosophilidae | *Drosophila cancellata* Mather | late instar | | | self-anchored loop, ventral | decaying fruit | [19–21] |
| | | *Drosophila coracina* Kikkawa | late instar | | | self-anchored loop, ventral | decaying fruit | [19, 22] |
| | | *Drosophila enigma* Malloch | late instar | | | self-anchored loop, ventral | decaying fruit | [19–21] |
| | | *Drosophila immigrans* Sturtevant | late instar | | | self-anchored loop, ventral | decaying fruit | [19] |
| | | *Drosophila lativittata* Malloch | late instar | | | self-anchored loop, ventral | decaying fruit | [19–21] |
| | | *Drosophila levis* Mather | late instar | | | self-anchored loop, ventral | decaying fruit | [19–21] |
| | | *Drosophila maculosa* Mather | late instar | | | self-anchored loop, ventral | decaying fruit | [19–21] |
| | | *Drosophila opaca* Mather | late instar | | | self-anchored loop, ventral | decaying fruit | [19–21] |
| | | *Drosophila subtilis* Kikkawa & Peng | late instar | | | self-anchored loop, ventral | decaying fruit | [19, 22] |
| | | *Scaptodrosophila kirki* (Harrison) | late instar | | | self-anchored loop, ventral | decaying fruit, fungus | [19] |
| | Lonchaeidae | *Dasiops caustonae* Norrbom & McAlpine | late instar | 100 mm | | self-anchored loop, ventral* | fresh fruit of *Passiflora mollissima* | [23] |
| | | *Dasiops vibrissata* (Malloch) | late instar | | | self-anchored loop, ventral | fungus under bark of dead tree | observations during this study |
| | | *Lonchaea filifera* Bezzi | late instar | | | self-anchored loop, ventral* | decaying organic matter | [17] |
| | Mycetophilidae | *Mycetophila cingulum* Meigen | last instar | 150 mm | ~1.0 m/s | self-anchored loop, dorsal | polypore, *Polyporus squamosus* | [12] |
| | Phoridae | *Chonocephalus depressus* De Meijere | last instar | | | self-anchored loop, ventral* | decaying organic matter | [24] |

*(Continued)*

**Table 1.** (Continued)

| Order | Family | Species | Life Stage | Maximum Distance | Speed | Mechanism | Host or substrate | Citation |
|---|---|---|---|---|---|---|---|---|
| | Piophilidae | *Piophila casei* (Linnaeus) | unknown | | | self-anchored loop, ventral | cheese | [7] |
| | | *Prochyliza xanthostoma* Walker | late instar | 500 mm | | self-anchored loop, ventral | carrion | [25] |
| | | *Stearibia nigriceps* (Meigen) | late instar | | | self-anchored loop, ventral* | carrion | [25] |
| | | *Liopiophila varipes* (Meigen) | late instar | | | self-anchored loop, ventral* | carrion | [25] |
| | | *Protopiophila latipes* (Meigen) | late instar | | | self-anchored loop, ventral* | carrion | [25] |
| | | *Parapiophila* spp. | late instar | | | self-anchored loop, ventral* | carrion | [25] |
| | Pipunculidae | *Pipunculus annulifemur* Brunetti** | last instar | | | unknown | endoparasite of Auchennorhyncha | [10, 26] |
| | Platystomatidae | *Scholastes aitapensis* Malloch | unknown | | | self-anchored loop, ventral* | decaying plant matter, dung | [17] |
| | Sepsidae | Unknown | unknown | | | self-anchored loop, ventral* | dung and decaying materials | [27] |
| | Tephritidae | *Ceratitis capitata* (Wiedemann) | last instar | 120 mm | 0.5 m/s | self-anchored loop, ventral* | fruit | [28] |
| | Ulidiidae | *Euxesta notata* Wiedemann | last instar | | | self-anchored loop, ventral* | decaying plant matter | [29] |
| | | *Notogramma cimiciforme* Loew (as *N. stigma*) | last instar | | | self-anchored loop, ventral* | decaying plant matter | [17] |
| Lepidoptera | Pyralidae | *Emporia melanobasis* Balinsky | last instar | | | Unknown | hollowed fruit | [30] |
| | Thyrididae | *Calindoea trifascialis* (Moore) | last instar | | | substrate-anchored loop | dipterocarp leaf | [31] |
| | Tortricidae | *Cydia saltitans* (Westwood) | last instar | | | substrate-anchored loop | hollowed seed | [32, 33] |
| Hymenoptera | Cynipidae | *Neuroterus saltatorius* Edwards | last instar larva | 30 mm | | Unknown | hollowed gall | [34, 35] |
| | Eucharitidae | *Dicoelothorax platycerus* Ashmead | first instar/planidium | | | substrate-anchored cercal spring | active among host habitats; feed on ant larvae | [36] |
| | | *Galearia latreillei* (Guérin-Méneville) | first instar/planidium | | | substrate-anchored cercal spring | active among host habitats; feed on ant larvae | [36] |
| | | *Latina rugosa* (Torréns, Heraty, & Fidalgo) | first instar/planidium | | | substrate-anchored cercal spring | active among host habitats; feed on ant larvae | [36] |
| | | *Neolirata alta* (Walker) | first instar/planidium | | | substrate-anchored cercal spring | active among host habitats; feed on ant larvae | [36] |
| | | *Neolirata daguerrei* (Gemignani) | first instar/planidium | | | substrate-anchored cercal spring | active among host habitats; feed on ant larvae | [36] |
| | Ichneumonidae | *Bathyplectes anurus* (Thomson) | last instar | 50 mm (vertically) | | substrate-anchored spring (?) | rigid cocoon; parasitoid of alfalfa weevil | [37, 38] |
| | Perilampidae | *Monacon robertsi* Boucek | first instar/planidium | | | substrate-anchored cercal spring | active among host habitat; feed on beetle pupa | [39] |

(*Continued*)

**Table 1.** (Continued)

| Order | Family | Species | Life Stage | Maximum Distance | Speed | Mechanism | Host or substrate | Citation |
|---|---|---|---|---|---|---|---|---|
| | Tenthredinidae | *Heterarthrus* spp. | last instar | | | Unknown | flexible cocoon of leaf tissue | [40] |
| Coleoptera | Brentidae | *Nanophyes* sp. | late instar | | | Unknown | *Tamarix* seed capsules | [41] |
| | Carabidae | *Cicindela duodecimguttata* Dejean | third instar | | | unanchored loop, dorsal flexion followed by ventral flexion | sand, soil | [6] |
| | | *Cicindela lengi* Horn | third instar | | | unanchored loop, dorsal flexion followed by ventral flexion | sand, soil | [6] |
| | | *Cicindela tranquebarica* Herbst | third instar | | | unanchored loop, dorsal flexion followed by ventral flexion | sand, soil | [6] |
| | | *Habroscelimorpha dorsalis* Say | third instar | | | unanchored loop | sand | [5] |
| | | *Omus dejeani* Reiche | third instar | | | unanchored loop, dorsal flexion followed by ventral flexion | sand, soil | [6] |
| | | *Tetracha carolina* (Linnaeus) | third instar | | | unanchored loop, dorsal flexion followed by ventral flexion | sand, soil | [6] |
| | Curculionidae | *Conotrachelus anaglypticus* (Say) | unknown | 89 mm | | self-anchored loop, ventral | under bark of wounded trees | [42] |
| | Laemophloeidae | *Laemophlous biguttatus* (Say) | late instars | 11.2 mm | 0.47 m/s | substrate-anchored loop | fungus under bark of dead tree | this study |
| | | *Placonotus testaceus* (F.) | unknown | | | substrate-anchored loop | fungus under bark of dead tree | this study |
| Strepsipstera | Corioxenidae | *Corioxenos* sp. | first instar/ planidium | | | substrate-anchored spring (?) | endoparasite of Hemiptera | [10] |
| | Mengenillidae | *Eoxenos laboulbeni* de Peyerimhoff | first instar/ planidium | | | substrate-anchored spring (?) | endoparasite of Lepismatidae | [10] |
| | Myrmecolacidae | *Stichotrema dallatorreanum* Hofeneder | first instar/ planidium | | | substrate-anchored spring (?) | endoparasite of Hymenoptera | [43, 44] |

*presumed based on other related taxa

**Skevington and Marshall [45] note that Subramaniam's observation of jumping *P. annulifemur* must have been another genus, as *Pipunculus* only parasitizes deltocephaline cicadellids.

kinematic performance such as acceleration and power output (e.g. jumping larvae of the gall midge *Asphondylia* sp. in Farley et al. [13]). These kinematic measures can then be used as a metric to determine if jumps can be explained as the result of direct muscle movement alone, or if additional components, such as a latch-mediated spring actuation mechanism (abbreviated "LaMSA"; [46]), are involved. Some LaMSA systems are known or thought to utilize hydrostatic body deformations or deformations of a cuticular spring such as resilin or a resilin composite material to amplify the power output of direct muscle action [13, 47–51]. Typically, latches involving a mechanical interaction of one or more body components are employed to mediate the storage and release of this energy [45]. However, our view of the diversity and

functionality of LaMSA systems in larval insects is limited, as most examples remain undescribed and unresolved at the necessary level of mechanical detail.

Here we report and describe the mechanics of the first observation of latch-mediated escape jumping in a beetle larva (Coleoptera: Laemophloeidae: *Laemophloeus biguttatus* (Say)), using a novel mechanism that does not involve the looped body formation observed in many jumping insect larvae and appears to use attachment to the substrate as an anchor or latch. We also report observations of a similar behavior in another laemophloeid beetle larva, *Placonotus testaceus* (F.), and present a brief review of jumping behaviors in insect larvae.

## Materials and methods

### Specimen collection

In October of 2019, beetle larvae were collected from under the bark of a standing, dead Darlington oak (*Quercus hemisphaerica* W. Bartram ex Willd.) exhibiting abundant growth of the fungus *Biscogniauxia atropunctata* (Schwein.) Pouzar (Fig 1). The tree was about twelve (12) inches (30.5 cm) DBH and located on the South side of Governors Scott Courtyard on the main campus of North Carolina State University (35˚47'15.0"N, 78˚40'23.7"W). Numerous beetle larvae and adults (Laemophloeidae, Monotomidae, Mycetophagidae, Latridiidae, and others), flies and their larvae (Lonchaeidae, Ulidiidae), flat bugs (Aradidae), mites (Astigmatina), termites (*Reticulitermes*), ants (Formicidae, including *Brachymyrmex* and *Solenopsis invicta* Buren), and other arthropods were present. Various live insects were collected by MAB for photos and to preserve specimens for the NC State University Insect Museum. The insects were brought into the lab in small covered containers with some of the removed bark and kept moist with a damp paper towel until photos could be taken.

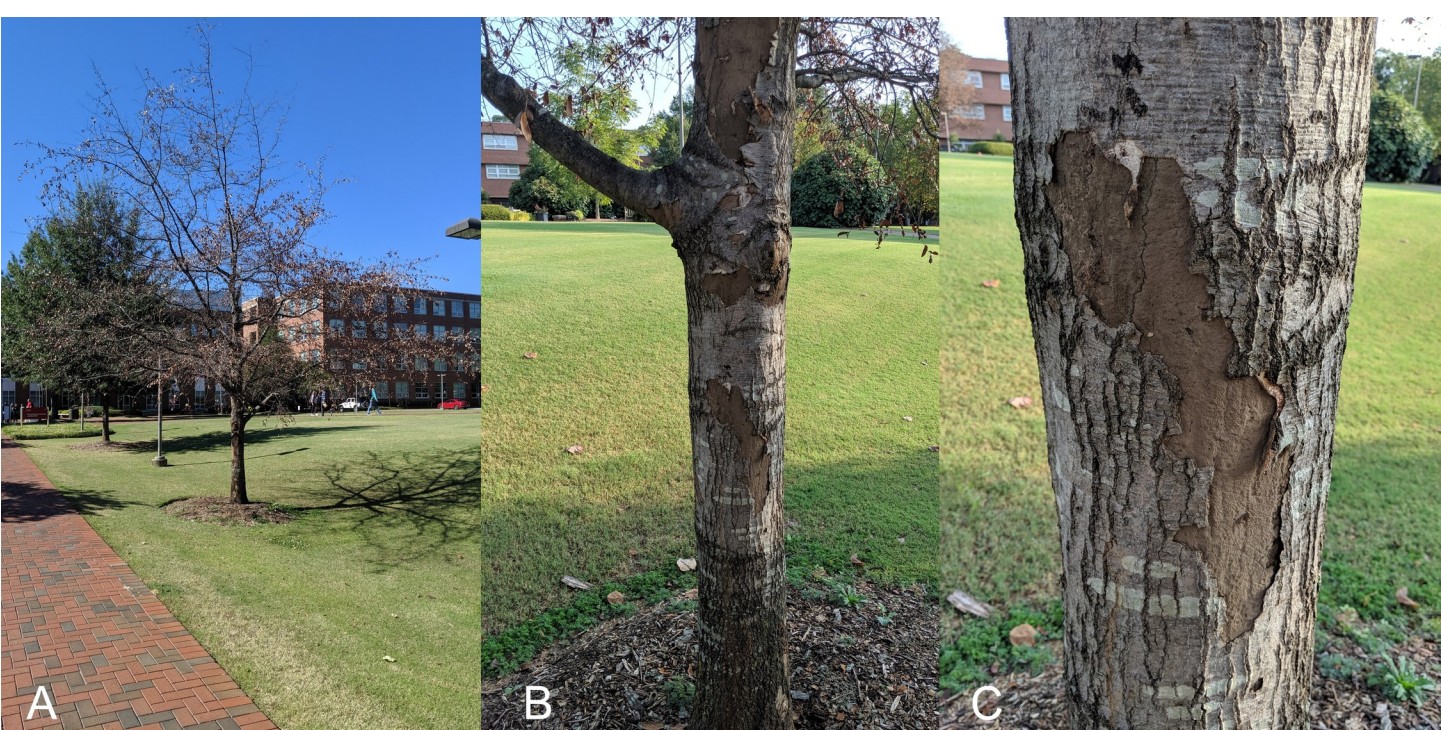

**Fig 1.** A: Location of the Darlington oak (*Quercus hemisphaerica* W. Bartram ex Willd.) exhibiting abundant growth of the fungus *Biscogniauxia atropunctata* (Schwein.) Pouzar where the beetles and larvae of *Laemophloeus biguttatus* were collected; B&C: Images showing the growth of the fungus and peeling bark (October of 2019; photos by MAB).

The larva of another cucujoid beetle was collected by TY from under the bark of a dead, broad-leaf tree with several conspecific adults and larvae on August 2nd, 2020. The tree had been cut down and lying in a place with good sunlight near the parking of Narukawa Valley, Kihokuchô Town, Ehime Pref., Japan (33˚13'02.9"N 132˚37'16.9"E). Several arthropods including non-laemophloeid beetles were also found associated with the dead tree.

## Larval identification

Larvae from the North Carolina site were presumed to be juveniles of one of the abundant beetle species associated with the tree, providing an initial starting point for identification. We used the Coleoptera keys in Stehr [52] to initially identify larvae to family level. For species ID we amplified a 591 bp section of the mitochondrial cytochrome c oxidase subunit I (CO1), from a single larva, for comparison with published sequences. Primers used for amplification of this gene were based on [53]. Voucher specimens of larvae are deposited in the NC State University Insect Museum (GBIF: http://grbio.org/cool/ij62-iybb).

Initially, the examined larva of the Japanese species was regarded as a laemophloeid based on overall morphological features: the flattened body, the well sclerotized, large, longitudinal abdominal segment VIII, and the small, well sclerotized, horn-like urogomphi [54, 55]. The species was identified based on the adult morphology after rearing the larva to the adult stage. The larva was transferred to a Petri dish (Becton Dickinson; diameter 50 mm, height 9 mm) for the rearing after transporting them into the laboratory. Some pieces of the bark of the dead tree, from which the larva was found, and a wet tissue squeezed tightly were placed in the dish to provide food for the larva and moisture. The dish was retained in the laboratory at room temperature until eclosion.

## Jumping behavior

To capture the jumping behavior for analysis, *L. biguttatus* larvae were placed on a 6cm x 2cm acrylic platform affixed to a backing board with a 0.5cm$^2$ scale grid. Because larvae first proved unable to perform jumps on the smooth bare acrylic (see results), a single layer of standard 20 lb thickness copy paper was glued (Elmer's All Purpose non-toxic, acid-free) to the surface of the platform. Jumps were filmed at a rate of 3,200 frames s$^{-1}$ at a pixel resolution of 1280 x 720 using a Phantom Miro LC321s (Vision Research, Wayne, New Jersey, US), through a 60mm f/ 2.8 2X Ultra-Macro lens (Venus Optics/Laowa, Hefei, Anhui, China). Images were gathered at a lens aperture setting of f/8-11, and at a magnification ratio of approximately 1:1. Image exposure time was 0.156 ms. The platform and insects were front-lit with an high-intensity LED light array (Visual Instrumentation Corporation, Lancaster, CA, US). The videos were captured at a frame wide enough to include the entire trajectory of the jumps. To confirm that 3,200 frames s$^{-1}$ was fast enough to resolve all jump-related rapid movements, an additional eight jumps from two individuals were filmed at 60,000 frames s$^{-1}$. To do so, we used a Photron FASTCAM SA-Z filming at a pixel resolution of 896 x 368 with an image exposure time of 1.05 μs under the same conditions as described above.

The larvae were placed on the platform, unrestrained and filmed continuously until they performed a jump. Beyond being exposed to intense lighting, the larvae were not prodded or stimulated to perform jumps in any way. For analysis we filmed 39 jumps across 12 individuals. Of those jumps filmed, 29 jumps from 11 of the 12 individuals were performed at an angle perpendicular to the camera, allowing us to perform the analyses described below. The remaining 10 jumps were excluded from additional analyses.

After all jumps were filmed, 15 beetle larvae (including the 12 that were filmed for analysis) were weighed using a balance sensitive to a tenth of a milligram (Denver Instruments PI-

114N). Due to the sensitivity of the balance used, average mass was used for all jump calculations, since mass values could not be associated with individual beetles.

The larva of *P. testaceus* was placed on a Petri dish (Becton Dickinson; diameter 50 mm, height 9 mm) covered with a piece of wet tissue paper at room temperature. The larval jumping behavior was filmed at a rate of 30 frames s$^{-1}$ at a pixel resolution of 1920 x 1080 using a digital camera (Canon EOS 7D) fitted with a macro-objective (MP-E 65 mm), while illuminating the platform using an LED light (Hayashi-repic, HDA-TW3A). The larval locomotion was tracked with the camera by hand until successfully capturing the jumping behavior.

## Video analysis

To examine the jumps of *L. biguttatus* under a power amplification and LaMSA framework, jumps were initially divided into four phases based on the movements, actions, and body positioning of the larval beetles: 1) an initial load phase (Fig 2A), which is thought to correspond to the contraction of muscles storing energy in the elastic components of the body, was characterized as starting when the larvae first begin to arch their body dorsally, and ending when the larvae's legs begin to lose contact with the ground; 2) a latch-decoupling phase (Fig 2B), which occurs as the larvae release their grip on the substrate, was characterized as starting immediately after the end of the loading phase and ending when the final leg loses contact with the ground, as based on initial observations of jumps and SEM images of larvae (see external and internal morphology subsection of the Results) the legs appeared to be the most likely candidates for a latching mechanism, if one is indeed present; 3) a launch phase (Fig 2C), which corresponds to the transfer of stored elastic energy within the bodies to kinetic energy of the jumps, was characterized as beginning during or after the end of the latch-decoupling phase and ends when all contact between the larvae's bodies and the substrate has ceased; and 4) an airborne phase (Fig 2D), which was characterized as beginning after the end of the launch phase and finishing when the larvae land. The frames where phase transitions occurred were manually recorded for each video and used for temporal calculations of the jumps' phases. However, after this initial characterization it became evident that the distinction between the latch-decoupling and launch phases as we defined them were not always easily discerned; substantial overlap between these two phases was often observed such that rapid curling of the body occurred when one or more legs were still in contact with the platform. As a result, these two phases were combined for later estimations of power density as explained below. Body rotation for each jump, while in the air, was manually estimated and reported in Table 2.

Tracking of the larvae's movement was completed in ImageJ ver 1.52a [56]. Videos were converted to 8-bit grayscale and thresholded to generate binary images. The movement of the larvae was then auto-tracked using the Multitracker plugin [57], which estimated the larvae's center of mass using the centroid of the converted images and traced movement of the centroid through each frame. The angles of the larvae's bodies at the end of the loading, latch-decoupling, and launch phases were measured using ImageJ's default angle tool.

The xy coordinates through time of each jump were imported into R ver 3.5.2 [58], where they were scaled using the 0.5cm$^2$ grid included in each video and reoriented so that each jump started at the origin of a cartesian grid and proceeded in the positive x and y directions. A parabola was fit to each trajectory using the *poly* function, and the maximum height (h) and horizontal distance (d) traveled over the course of the airborne phase of each jump were calculated as the y coordinate of the vertex and positive x-intercept, respectively, using the following equations:

$$h = \frac{-b}{2a} \tag{1}$$

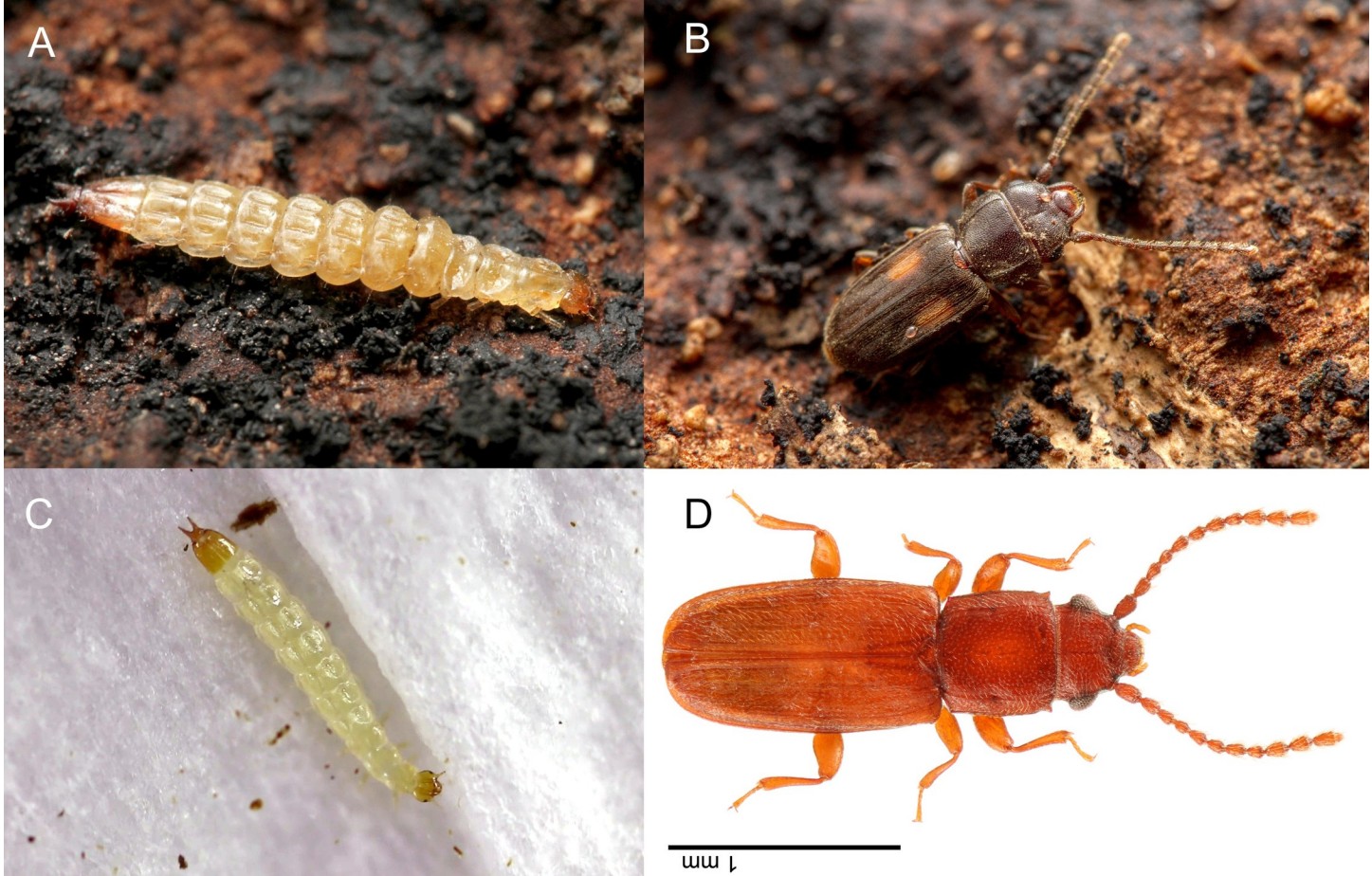

**Fig 2. *Laemophloeus biguttatus* jump sequences (A-C and D are separate jumps) taken from videos filmed at 3,200 frames per second.** A-C: loading, latch-decoupling, and launch phases of a jump, timecode labels on images correspond to the image of the beetle they are nearest; D: complete jump trajectory. A: loading phase, during which the body slowly bends ventrally. 0.119 seconds elapse between top and bottom body postures. Bottom image is the frame directly preceding the top larval image in B. B: latch-decoupling sequence, during which the legs release or lose their grip on the substrate. In the pictured jump sequence, the hind and midlegs are first to release their grip, followed by the forelegs. Each larval image is a single sequential frame and only 0.625 ms separate the top and bottom image. C: launch phase, corresponding to the transfer of stored energy to the kinetic energy of the body moving into the air. Shown here are the last frames of the launch phase depicting the last frame in which the larvae has any contact with the substrate (below) and a frame from the airborne phase (above). 5.31 ms separate the bottom image from the top. The bottom image in panel C is 1.8 ms after the bottom image in panel B. D: Airborne phase of a separate jump from that depicted in panels A-C. The entire sequence spans 0.081 seconds, noted times pertain to the first, top, and last of the sequential images, and the scale bar pertains only to this panel.

$$d = \frac{-b \pm \sqrt{b^2 - 4ac}}{2a} \tag{2}$$

Where a, b, and c correspond to the coefficients of the fitted parabolic equation in the form of $y = ax^2+bx+c$. The takeoff angle ($\alpha$) was calculated as:

$$\alpha = tan^{-1}(b) \tag{3}$$

Cumulative displacement at each time point was calculated, and the amount of displacement occurring during each of the pre-airborne phases was estimated by dividing the cumulative displacement between the four phases based on the frames at which transitions between phases as defined above occurred. Displacement occurring during the latch-decoupling and launch

**Table 2. Jump kinematics based on high-speed imaging from several larvae of *Laemophloeus biguttatus*.**

| No. of jumps | Body Length (x 10-3 m) | Est. body Mass (x 10-6 kg) | Loading phase (sec) | Latch release phase (x 10-3 sec) | Launch phase (x 10-3 sec) | Launch phase distance (x 10-3 m) | Avg. mass-specific power of launch (W kg-1) | Max takeoff speed (m s-1) | Kinetic energy at takeoff (x 10-7 J) | Takeoff angle (deg) | Total jump pitch (+/- %) | Total jump roll (+/- %) | Total jump yaw (+/- %) | Jump horizontal distance* (x 10-3 m) | Max jump height (x 10-3 m) |
|---|---|---|---|---|---|---|---|---|---|---|---|---|---|---|---|
| 4 | 6.22 | 1.3 | 0.12 | 3.6 | 1.48 | 1 | 118 | 0.39 | 1.03 | 106.4 | 50 | 15.6 | 28.3 | 11.8 | 5.9 |
| | | | (0.05-0.16) | (1.9-5.9) | (0.31-3.44) | (0.8-1.4) | (44-258) | (0.33-0.48) | (0.73-1.51) | (65.8-146.4) | (0-100) | (-37.5-125) | (-25-138) | (6.5-17.5) | (2.3-13) |
| 4 | 5.94 | 1.3 | 0.33 | 21.1 | 1.09 | 1.2 | 334 | 0.51 | 1.7 | 83.7 | -37.5 | -93.75 | -37.5 | 14 | 13.2 |
| | | | (0.26-0.41) | (1.3-79.7) | (0.94-1.25) | (0.6-2.4) | (0.11-693) | (0.44-0.54) | (1.25-1.91) | (49.9-104) | (-150-0) | (-200-0) | (-150-0) | (5.9-19.7) | (11.5-14.8) |
| 2 | 5.46 | 1.3 | 0.27 | 2 | 1.72 | 1.1 | 229 | 0.39 | 0.99 | 95 | -50 | -12.5 | 0 | 7.5 | 8.5 |
| | | | (0.25-0.3) | (1.9-2.2) | (1.56-1.88) | (1-1.2) | (202-257) | (0.36-0.42) | (0.84-1.15) | (93.9-96.2) | (-100-0) | (-50-25) | (0-0) | (5.6-9.4) | (7.1-9.9) |
| 2 | 6.32 | 1.3 | 0.14 | 3.9 | 1.56 | 1.3 | 104 | 0.29 | 0.53 | 77.5 | n/a** | n/a | n/a | 7.9 | 3.9 |
| | | | (0.11-0.17) | (2.8-5) | (0.63-2.5) | (1.3-1.3) | (99-108) | (0.28-0.29) | (0.51-0.55) | (64.1-90.9) | | | | (7.3-8.5) | (3.1-4.8) |
| 4 | 5.53 | 1.3 | 0.51 | 3.2 | 0.86 | 0.9 | 159 | 0.32 | 0.78 | 56 | 25 | -18.75 | 70.75 | 10.3 | 4.6 |
| | | | (0.23-0.68) | (2.2-4.7) | (0.31-1.88) | (0.3-1.2) | (34-411) | (0.14-0.51) | (0.12-1.69) | (14.3-87.2) | (0-100) | (-50-0) | (0-183) | (2.3-23.7) | (1.9-7.4) |
| 3 | 5.86 | 1.3 | 0.1 | 1.1 | 1.46 | 1.3 | 913 | 0.59 | 2.26 | 88.9 | -33.33 | -12.5 | -8.33 | 13.6 | 7.3 |
| | | | (0.03-0.16) | (0.9-1.3) | (1.25-1.56) | (0.9-1.6) | (545-1206) | (0.5-0.63) | (1.64-2.62) | (68.7-100.4) | (-50-0) | (-37.5-0) | (-25-0) | (12.6-14.3) | (3.1-9.8) |
| 1 | 4.24 | 1.3 | 0.3 | 8.8 | 0.63 | 1.4 | 24 | 0.54 | 1.90 | 62.1 | 0 | -75 | -50 | 11.8 | 7.6 |
| 1 | 4.88 | 1.3 | 0.15 | 0.6 | 1.56 | 0.9 | 767 | 0.55 | 1.99 | 73.3 | -50 | 37.5 | 0 | 8.7 | 8.5 |
| 3 | 5.22 | 1.3 | 0.1 | 3.2 | 1.77 | 1.2 | 121 | 0.46 | 1.38 | 74.5 | 0 | 16.67 | -50 | 11.1 | 11.1 |
| | | | (0.05-0.14) | (2.2-4.1) | (1.25-2.19) | (1-1.5) | (74-157) | (0.4-0.52) | (1.07-1.75) | (65-81.8) | (0-0) | (-50-50) | (-100-0) | (9.4-14) | (8.6-14.2) |
| 3 | 5.82 | 1.3 | 0.11 | 2.9 | 1.77 | 2.5 | 642 | 0.72 | 3.62 | 95.6 | 16.67 | 0 | -41.67 | 14 | 3.6 |
| | | | (0.06-0.17) | (1.9-4.7) | (0.31-2.81) | (1.3-4) | (259-1319) | (0.47-0.87) | (1.42-4.89) | (74.3-109.6) | (-100-100) | (-25-25) | (-100-0) | (12.1-15.5) | (1.5-7.7) |
| 2 | 5.58 | 1.3 | 0.13 | 3 | 2.19 | 2.1 | 213 | 0.63 | 2.74 | 58.9 | -112.5 | 0 | 0 | 9.8 | 11.2 |
| | | | (0.09-0.17) | (1.3-4.7) | (1.56-2.81) | (1.1-4) | (119-307) | (0.5-0.87) | (1.66-4.89) | (27.9-109.6) | (-125–100) | (0-0) | (0-0) | (4-15.5) | (7.7-13.2) |
| Avg ±SD | 5.55 ±0.6 | 1.3 | 0.22 ±0.17 | 5.5 ±14.4 | 1.4±0.8 | 1.3±0.7 | 323±353 | 0.47 ±0.15 | 1.59 ±1.05 | 79.6 ±28.2 | -10.2 ±66.6 | -16.2 ±63.4 | -3.9 ±65.5 | 11.2±4.8 | 7.9±4.3 |
| Range | | | (0.03-0.68) | (0.63-79.69) | (0.31-3.44) | (0.32-4.02) | (0.11-1319) | (0.14-0.87) | (0.12-4.89) | (14.3-146.4) | (-150-100) | (-200-125) | (-150-183) | (2.33-23.75) | (1.51-14.83) |

Each row corresponds to jumps of individual larvae, with averages and ranges for all jumps from all larvae included in the final two rows.

* jumps were filmed from a single angle, so not all jumps were perfectly parallel to the plane of view and horizontal distance estimates may, therefore, be underestimated.

** rotational data unavailable as beetle collided with the wall on descent, altering normal body rotation.

phases were combined for calculations of power and power density, as the latch-decoupling phase as we characterized it varied widely in its duration and sometimes encompassed the entirety of the launch phase, and a non-negligible elevation of the larvae's centers of mass sometimes occurred during the latch-decoupling phase in these jumps. As this could be interpreted as either an indication that the presence or identity of the latching mechanism was misidentified, or that the legs serve as imperfect latches, we proceeded assuming the latter instance. While probably not ideal, pooling of these two phases for this calculation added a

conservative bias to estimates of power and power density. A spline function was fitted to the cumulative displacement data using the *smooth.pspline* function from the pspline package in R with a smoothing spar value of $10^{-8}$ [59]. This spar value was visually determined to fit the datasets sufficiently while not resulting in exceptionally noisy derivative curves. Velocity and acceleration curves were calculated by taking the first and second derivatives of the displacement splines, respectively, and takeoff velocities and accelerations were estimated as the maximum values of these curves. Jump energy (E) was calculated as:

$$E = 0.5mv^2 \qquad (4)$$

Where m is the body mass of the beetle and v is the takeoff velocity. Jump power (P) was calculated as:

$$P = mL^2t^{-3} \qquad (5)$$

Where L is displacement of the center of mass attributable to the latch-decoupling and launch phases and t is the combined duration of the latch-decoupling and launch phases. Maximum average power density/ output, (O) during the latch-decoupling and launch phase was estimated by assuming that a certain proportion of the beetles' body mass was contributing to energy input during this phase:

$$O = Pm^{-1}c^{-1} \qquad (6)$$

Where c is the assumed proportion of the beetles' body mass powering jumps. As the exact jumping mechanism (and the muscles powering it) is unknown, an upper bound for this value was estimated by measuring the total volume of all muscle within the beetle's body via microCT data (see below), and multiplying this by an assumed muscle density of 1060 kg m$^{-3}$, a value measured from mammalian muscle that has been previously used in calculations on insect muscle power [60–62]. This calculation revealed an estimated 9.78% of the beetles' total body mass to be composed of muscle. In addition to this estimate of c, power output calculations were also performed assuming 100%, 75%, 50%, 32.31%, 19.60% and 4.89% of the beetles' body mass were powering jumps, to account for potential errors in muscle measurements from the CT data due to shrinkage, the likely possibility that not all of the beetles' muscles are powering jumps, and to calculate the maximum percentage of the beetles' body mass that can be powering jumps and still not be able to explain the power output of at least one of the measured jumps. For each set of power density calculations the estimates were compared to the highest known value of maximum average power density produced from muscle (approximately 400 W kg$^{-1}$; [63]) to determine whether muscle contraction without a spring-latch system could feasibly produce the performances measured.

To compare locomotory performance of jumping vs crawling in beetle larvae the energetic cost of transport (COT$_{jump}$) of jumps was calculated following the methods of Farley et al. [13]:

$$COT_{jump} = \frac{10E}{md} \qquad (7)$$

This assumes an energetic efficiency of 10% for the muscles powering the jumps and has units of J kg$^{-1}$ m$^{-1}$ (i.e. the amount of energy required to move one kilogram of the beetle's body mass one meter). COT for crawling was estimated by substituting the average beetle mass into the power regression equation determined for body mass vs crawling COT for legged

arthropods by Full [64]:

$$COT_{crawl} = 10.8m^{-0.32} \tag{8}$$

Where m is the average larval body mass in kilograms.

Uncertainty ratios for velocity, acceleration, energy and power were calculated using formulas provided in Longo et al. [46]. To reduce uncertainty attributable to length measurements, the $0.5cm^2$ scale grid was measured more precisely using images of the grid taken with a Keyence VHX 5000 microscope and measured to a precision of 0.001mm. These calculations resulted in an average uncertainty of 8% for velocity, 16% for acceleration, 11% for kinetic energy, 26% for power, and 24% for mass-specific power density.

## MicroCT

To estimate muscle mass for power density calculations, as well as examine internal morphology to uncover the mechanism powering the jumps, one *L. biguttatus* larva was scanned using microCT at the Imaging Technology Group, Beckman Institute for Advanced Science and Technology, University of Illinois at Urbana-Champaign. This specimen was killed by being placed directly into Brasil fixative (Electron Microscope Company, Hatfield, PA) and left for 24 hours. The larva was then washed several times with 70% ethanol to remove excess fixative and taken through an ethanol series to 100% ethanol (1 hour each at 80%, 90%, 95% and 100%). To improve contrast between the cuticle and muscle tissue the larva was stained overnight in I2E immediately prior to scanning (1% iodine in 100% ethanol) and then washed several times in 100% ethanol the following morning. The specimen was dried using an AutoSamdri-931.GL Supercritical Point Dryer (Tousimis Research Corporation, Rockville, MD) and scanned using an Xradia MicroXCT-400 scanner (Carl Zeiss, Oberkochen, Germany) with power settings of 25kV voltage and 5W power. 1441 images were taken at an exposure time of 6 sec spanning a 360° view of the larva. A 4x lens was used and source and detector distances from the specimen were 59.1 mm and 15 mm, respectively. All reconstructions and segmentations were done in Amira ver 5.4.5 (FEI, Hillsboro, OR). In this program muscles were identified based on their shape and location within the scan, in addition to their increased contrast compared to other anatomical structures as a result of I2E staining, and segmented. The MaterialStatistics function was then used to calculate total volume of the segmented muscle.

## SEM

Three *L. biguttatus* larvae that were observed jumping were preserved and prepared for imaging through the following sequence. First, the live larvae were killed by a one-minute soak in boiling water. Next, they were transferred into 70% ethanol and stored for two weeks. Following this, the larvae were taken to the point of complete dehydration with 24-hour changes of room temperature 95% ethanol and three 100% ethanol changes. The larvae were then critical point dried in liquid $CO_2$ for 15 minutes at equilibrium using a Tousimis Samdri-795 critical point dryer (Tousimis Research Corporation, Rockville MD) and then mounted on stubs with double-stick tape and careful application of silver paint to help prevent charging in the microscope. Larvae were sputter coated with approximately 50Å of gold-palladium in a Hummer 6.2 sputtering system (Anatech USA, Hayward CA). Larvae were imaged using a JEOL JSM-5900LV at 10kV. Close inspection was done on parts of the body observed to make contact with the ground during the loading, latch, and launch phases of the jumps (specifically the ventral side of the head, the tarsi, and the ventral aspects of the terminal sections of the abdomen) to look for any potential morphological adaptations such as modified tarsal claws, friction

patches, or modified setae that might aid the larvae in adhering to the ground during the loading and latch phases of jumps.

## Results

### Identity of larvae

The larvae collected in North Carolina, USA (Fig 3A) were initially identified morphologically as Laemophloeidae based on their anatomy and the abundance of adults (*Laemophloeus biguttatus* (Say); Fig 3B) associated with the fungus. Identification was furthered by comparison to images from Bugguide.net (e.g. https://bugguide.net/node/view/241687/bgimage). A closer examination of morphology (including mouthpart dissection) and keys in Stehr [52] confirmed that the larvae belonged to Laemophloeidae (as Cucujidae: Laemophloeinae in that reference). Comparison of the larval CO1 sequence (GenBank: OK350080) to published sequences (NCBI Blast) resulted in a closest match (100%) with *L. biguttatus* (GenBank: KP134159).

The larva of the Japanese laemophloeid (Fig 3C) was successfully reared to adulthood (Fig 3D). The larva pupated on September 14th, 2020, and emerged on September 20th, 2020. The adult was conspecific with the laemophloeid adults collected with the larva and identified as *Placonotus testaceus* by comparison with published descriptions [65–67].

### Initial observations of jumps

After collecting larvae from their habitat, *L. biguttatus* specimens were brought into the lab to photograph under fluorescent lighting and room temperature conditions. Placing larvae on bark collected from the larval site, MAB noticed that they would rapidly crawl a short distance before jumping a short distance (S1 Video). Jumps were not instantaneous; instead, prior to jumping, the larvae stopped running or walking and pressed the anterior portion of their head (the mouthparts in particular) and pygidial region against the substrate. Abdominal segments

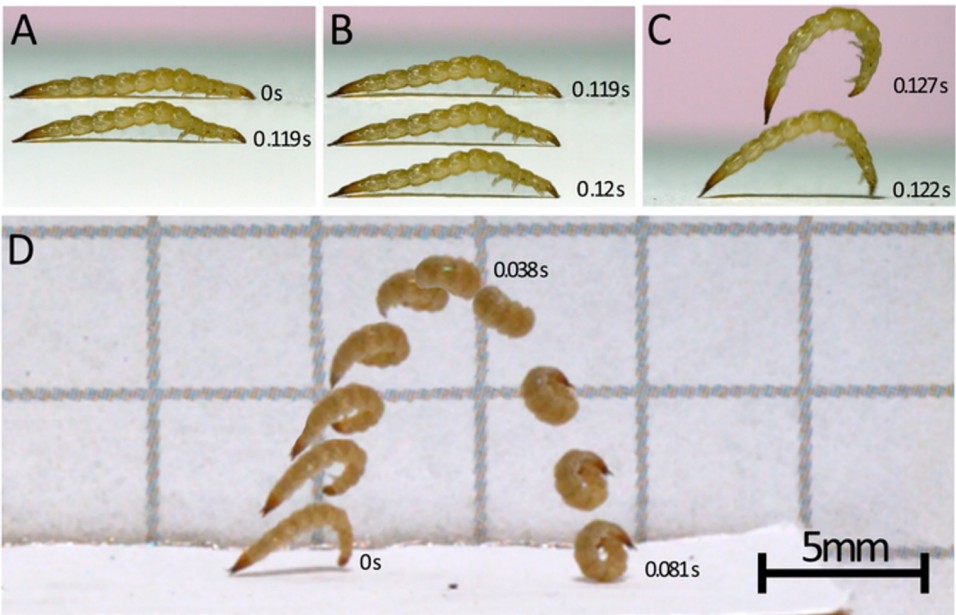

**Fig 3.** Habitus images of known Laemophloeidae with jumping larvae: A: larva of *Laemophloeus biguttatus*; B: same, adult; C: larva of *Placonotus testaceus*, D: same, adult. (A&B: taken by MAB; D&C: taken by TY).

1-6 were then arched up off the substrate while keeping the distal portion of the abdomen and the urogomphi in contact with the ground. From that posture, they rapidly curled their bodies ventrally into a jump (Fig 2, S2 Video). The larvae remain curled, in a complete loop, for the entirety of their jumps. After making impact with the ground in the curled posture, larvae bounced and rolled (if they did not land on their side) before uncurling and resuming leg-powered movement. Our further analysis of this behavior does not include post-jump rolling locomotion, which, under the right conditions, might also add to the total distance traveled during this behavioral sequence (e.g. [6]). The initial observations of these behaviors, across several individuals, was the motivation for pursuing slow motion video capture of the larvae.

During observation of the larva of *P. testaceus*, four jumps were filmed. As in the jumping behavior of *L. biguttatus*, the larva also crawled a short distance and took a posture flattening the head and distal abdominal segments against the substrate before each jump. In addition, at least three of the filmed jumping behaviors were observed just after the larva was dropped from a thin brush used for placing the larva on the platform. Although distances of all *Placonotus* jumps were not measured in detail, the longest jumping distance was about 5 cm, horizontally.

In preparation for slow motion video capture, *L. biguttatus* larvae were placed on smooth glass and acrylic platforms to test the suitability of each type as sets for video capture. On those substrates the larvae appeared to be unable to perform their jumps. Instead of jumping, larvae would struggle to grip the ground, and attempts to arch their abdomen or ventrally-curl their bodies into the jumping posture would result in toppling onto their side and back. Successful jumps off of these smooth surfaces were never observed. This observation, combined with high-speed video observations that show jumps starting when the legs lose their grip of the substrate (detailed below,) suggests that the larvae need to be able to anchor to themselves to the ground with their legs to build and release the energy for a jump.

## Jump performance

A summary of our analysis of high-speed video recordings of *L. biguttatus* jumps (n=11 larvae, 29 jumps total) is included in Table 2. Jump sequences began when the larvae stopped walking and arched their abdominal segments off the substrate (as described above) in a 'loading phase' which averaged $0.22 \pm 0.17$ s (mean ± standard deviation), and resulted in a change in body angle (head-to-posterior) from near horizontal to $149.5 \pm 16.7$ degrees. From this arched stance, the rapid ventral curling of their body was initiated when their tarsal claws slipped or were released and lost grip with the substrate (S2 Video). In all jumps where there was a clear view of the legs, the legs did not lose contact with the ground all at once; instead there was a 'latch-decoupling phase' between the first leg movement and the point at which the last legs left the ground averaging $5.5 \pm 14.4 \times 10^{-3}$ s in duration. In 26 of the 29 analyzed jumps, the larva was angled so that the front, middle, and hind legs were visible during this period. In 23 of those jumps the front legs were the last to lose contact with the ground, two sequences had a combination of middle and front legs leaving the ground last, and one had middle legs losing contact last. In addition to the 29 jumps we filmed at 3,200 frames per second, we captured eight jumps at 60,000 frames per second in order to verify that there were no other rapid movements that set the latch release phase in motion, preceding the legs losing contact. These sequences confirmed that tarsal claws losing grip with the ground is the first observable motion in the latch-decoupling phase (S2 Video). During the latch-decoupling phase the body of the larvae continues to arch further to $124.1 \pm 29.6$ degrees. The launch phase, or the time from when all legs have released to when all contact between the body of the larvae and the substrate is gone, averaged $1.4 \pm 0.8 \times 10^{-3}$ s. This phase corresponded to the elastic energy

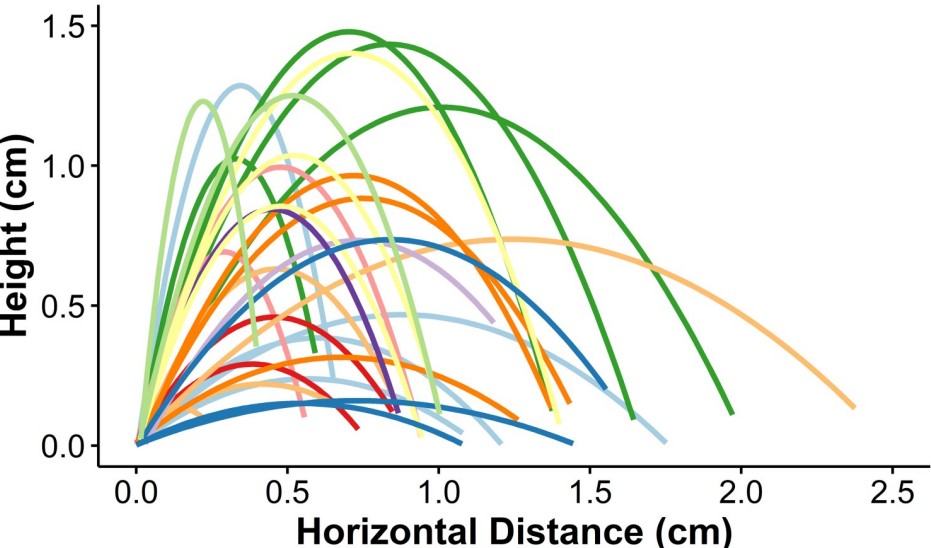

**Fig 4. Trajectories of all observed jumps of *L. biguttatus*.** Trajectories that share colors correspond to different jumps of the same larva.

stored within the body being transferred to kinetic energy of the body jumping off of the substrate. Note that this phase as defined above does not necessarily encompass the entire duration over which the larvae are accelerating, as larvae often began to accelerate during the latch-decoupling phase when only some of the legs had lost contact with the substrate, and in some jumps the last leg did not lose contact with the substrate until both the head and urogomphi lost contact as well. To ensure correct calculations of power and power density were made for jumps where this distinction is problematic, both the latch-decoupling phase and the launch phase as defined by leg positioning were combined so that the entirety of the period over which the larvae were accelerating was used. When pooled, the combined latch-decoupling and launch phases lasted 6.9 ± 14.3 ms. During the launch phase as defined by leg positioning, the larvae rapidly arched their body even further to 79.6 ± 28.2 degrees prior to takeoff, reached a maximum acceleration of 89.5 ± 34.5 m s$^{-2}$ and achieved a takeoff velocity of 0.47 ± 0.15 m s$^{-1}$ with the fastest takeoff velocity reaching 0.87 m s$^{-1}$, leaving the ground at an angle of 79.6 ± 28.2 degrees. Over the course of the jump larvae were airborne for 1.3 ± 0.7 x 10$^{-3}$ s and covered distances of 11.2 ± 2.8 mm horizontally and 7.9 ± 4.3 mm vertically, equivalent to 1.98 ± 0.8 and 1.5 ± 0.9 body lengths, respectively, though jump trajectories were variable with the farthest horizontal jumper traveling 23.75 mm (Fig 4). While airborne, three dimensional body rotation was minimal and is noted in Table 2. A cumulative displacement, velocity, and acceleration vs time plot for a representative jump is shown in Fig 5.

Results from the microCT scan revealed a total muscle volume of 0.12 mm$^3$ in the specimen examined (Fig 6A–6C). This volume had an estimated muscle mass of 0.12 mg, 9.78% of the average total mass of the beetle larvae filmed. Assuming that all of this muscle mass is used to power jumps (likely an overestimate), the maximum average power density during the launch phase of jumps was 323 ± 353 W/kg muscle, with a maximum of 1319 W/kg muscle (Fig 6D). Five of the 11 larvae filmed had at least one jump with an estimated maximum average power density exceeding the maximum recorded average power density for any muscle (400 W/kg), and three of those five had average power densities exceeding this value (Table 2). If only half of the total muscle mass (4.89% total body mass) was powering jumps, then eight of the 11 larvae had at least one jump with a maximum average power density that exceeded the 400W/kg

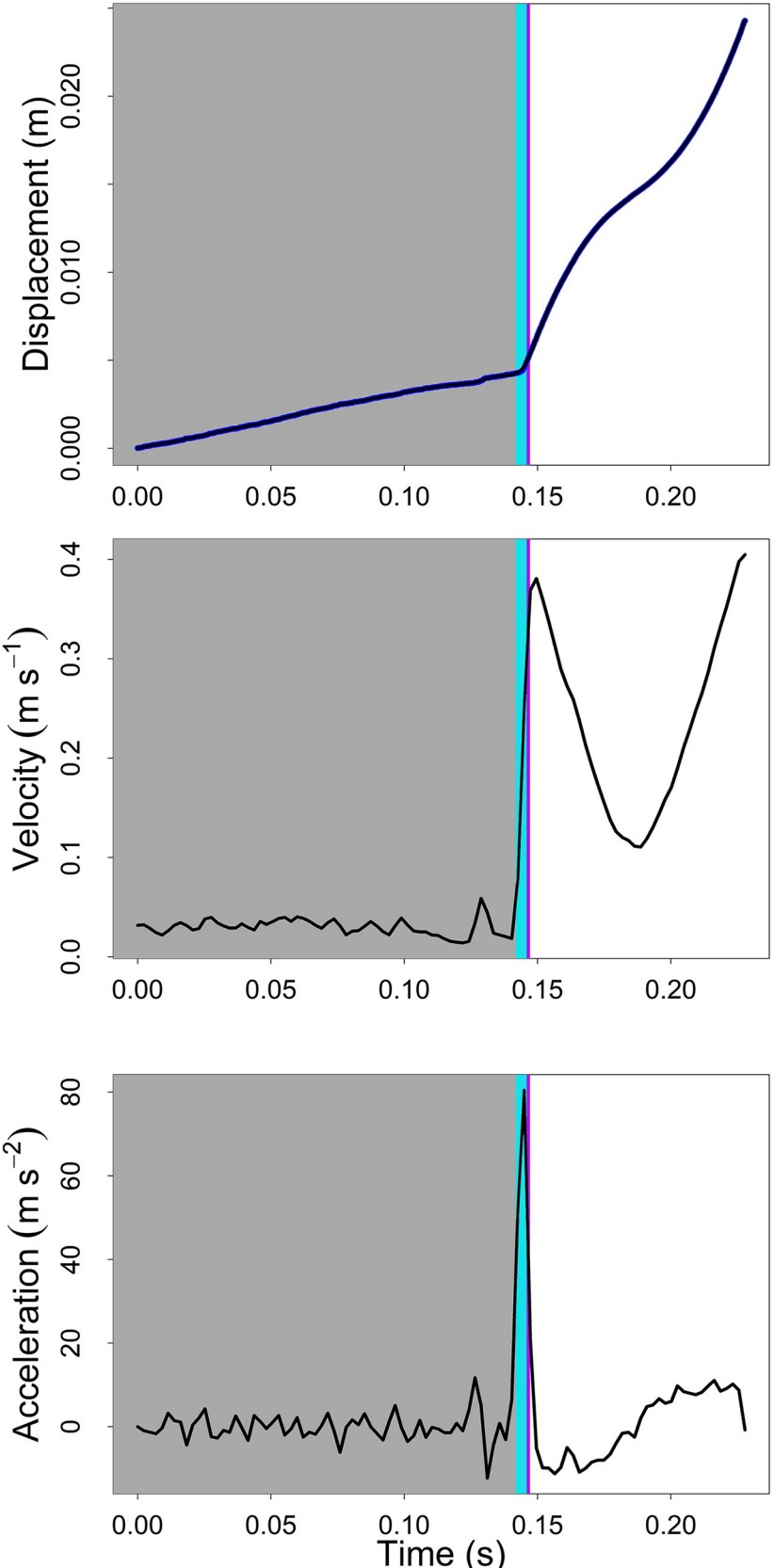

**Fig 5. Kinematic measurements of the jump of a beetle pictured in Fig 3D.** Loading phase is shown in grey, latch-decoupling phase shown in light blue, and launch phase ending when the beetle loses all contact with the ground is shown in purple. Dark blue on the displacement graph denotes actual data points while the black line represents the fitted spline function.

threshold (Fig 6D). It is possible that total percent muscle mass may be underestimated due to shrinkage occurring during the fixation process, so additional calculations of power density were done assuming 50% shrinkage of muscle (19.6% body mass composed of muscle). This still resulted in three jumps from two larvae having maximum average power densities exceeding the 400 W/kg threshold (Fig 6D). Only when the estimated muscle mass exceeded 32.31% of total body mass were power densities estimates of all jumps below the 400 W/kg threshold (Fig 6D).

The average energetic cost of transport for jumping ($COT_{jump}$) across all jumps was $110 \pm 74$ J $kg^{-1}$ $m^{-1}$, compared to an estimated cost of transport for crawling ($COT_{crawl}$) of 825 J $kg^{-1}$ $m^{-1}$ based on the power regression function for crawling arthropods calculated by Full [64].

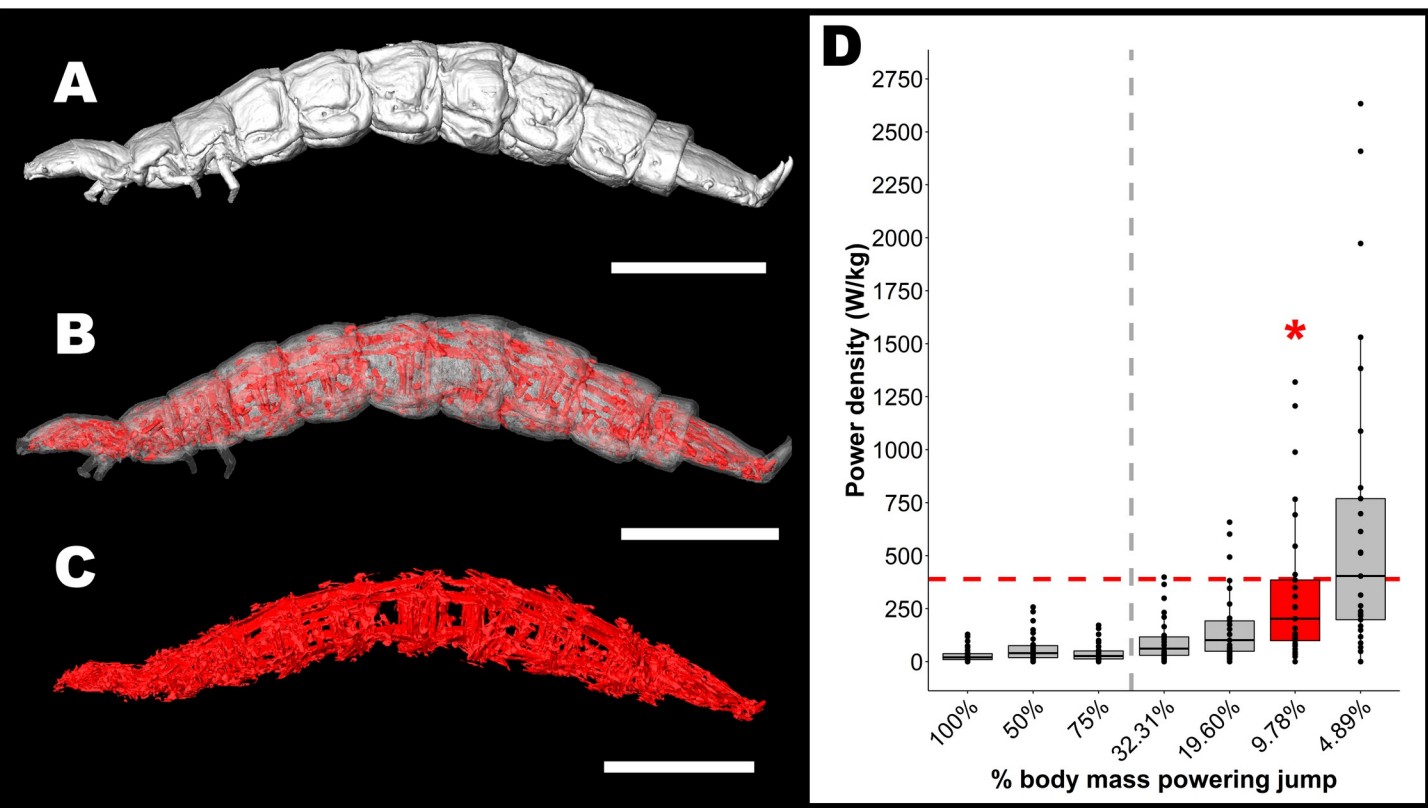

**Fig 6. Estimate of the contribution of muscle power in *L. biguttatus* jumps and evidence of a power amplification system.** Panel A-C: MicroCT whole-body imaging and isolation of muscles throughout the body cavity used to calculate total muscle mass. Scale bars denote 1mm. Panel D: power density (W/kg muscle) of jumps assuming differing proportions of the beetles' total body mass is being used to power jumps. Total body muscle mass was estimated to be 9.78% of the beetle's total body mass based on microCT data. At that mass estimate, using an overestimate that all of beetles muscles are involved in powering a jump, the power density for 7 of the 29 jumps we analyzed are beyond what can be explained by direct muscle contraction alone (those that are above the red dashed line), indicating the involvement of a power amplification mechanism. The red dashed line is reference to the 400 W $kg^{-1}$ high-power capability of vertebrate flight muscle [63]. If the muscles powering the jumps constitute more than 32.31% of the total body weight (left of the grey dashed line), then all analyzed jumps can be explained by direct muscle contraction alone.

### External and internal morphology (SEM/MicroCT)

SEM imaging of all body parts that were in direct contact with the substrate immediately prior to a jump did not reveal evidence of any micro- or nano-scale anatomical features which might be helping the larvae attach to the substrate during the loading phase of a jump (Fig 7). Likewise, the microCT scan revealed muscle arrangements similar to those of other insect larvae [68], with the musculature of the abdominal segments consisting of a series of dorso-ventral, dorsal longitudinal, and ventral longitudinal fibers (Fig 6A–6C). There did not appear to be any noticeable differences between abdominal segments in this arrangement.

### Review of jumping behavior in insect larvae

An extensive review of the literature was conducted in order to determine how common jumping behavior is within insect larvae; the results of this review are summarized in Table 1. Most authors provided a qualitative description of the jumping behavior without quantitative measurements of jump performance, but it is clear that some form of larval jumping is widespread in insects. This type of locomotion appears to have evolved in at least five orders of insects (as well as nematodes, not summarized), and is documented from at least 28 families, including the Laemophloeidae described herein. Given the phylogenetic distribution of jumping across unrelated orders and families, this behavior no doubt evolved repeatedly within holometabolous insects.

## Discussion

### Likelihood of power amplification and latch-mediated spring actuation

The results of our power density calculations for jumps provide a reasonable case for direct muscle action alone being insufficient to explain jump power for these larvae. Although the majority of jumps fall beneath our established 400 W/kg cutoff point for power amplification in all scenarios examined, this cutoff point is based on measurements from muscles that have been naturally selected for extraordinarily high sustained power output (bird flight muscle; [63]), and it is unlikely that actual power output of the larvae's muscles are that high. Additionally, combining the latch-decoupling and launch phases for power calculations conservatively biased our estimates towards lower power densities, since the latch-decoupling phase did not always heavily overlap with the launch phase for all jumps examined, though in instances where no overlap was observed the latch-decoupling phase was brief. Finally, as the exact spring mechanism and the associated muscles that power the jump are currently unknown, our estimations of muscle mass for power density calculations are certainly overestimates, further biasing our power density towards conservatively low values. Even with these conservative estimates, the fact that a non-negligible number (24%) of observed jumps had maximum average power densities exceeding the 400 W/kg threshold strongly suggests that direct muscle action alone is not responsible for powering jumps in all observed jumps.

If power amplification is indeed occurring in these beetle larvae, the low estimates of power density reported here compared to other power amplifying organisms could be a result of an imperfect latching system in which a substantial amount of energy is lost in jumps where the legs lose contact over an extended latch-decoupling phase, as latch decoupling time has been shown to have a substantial effect on energy flow and loss through LaMSA systems [69]. Alternatively, as morphological examination of the larvae did not reveal any obvious spring component, it is also possible that power is being amplified solely via direct muscle actuation accompanied by latch mediation, effectively forming a 'LaMMA (Latch-mediated muscle actuated)' system. If this is indeed the case then to the best of the authors' knowledge this would be

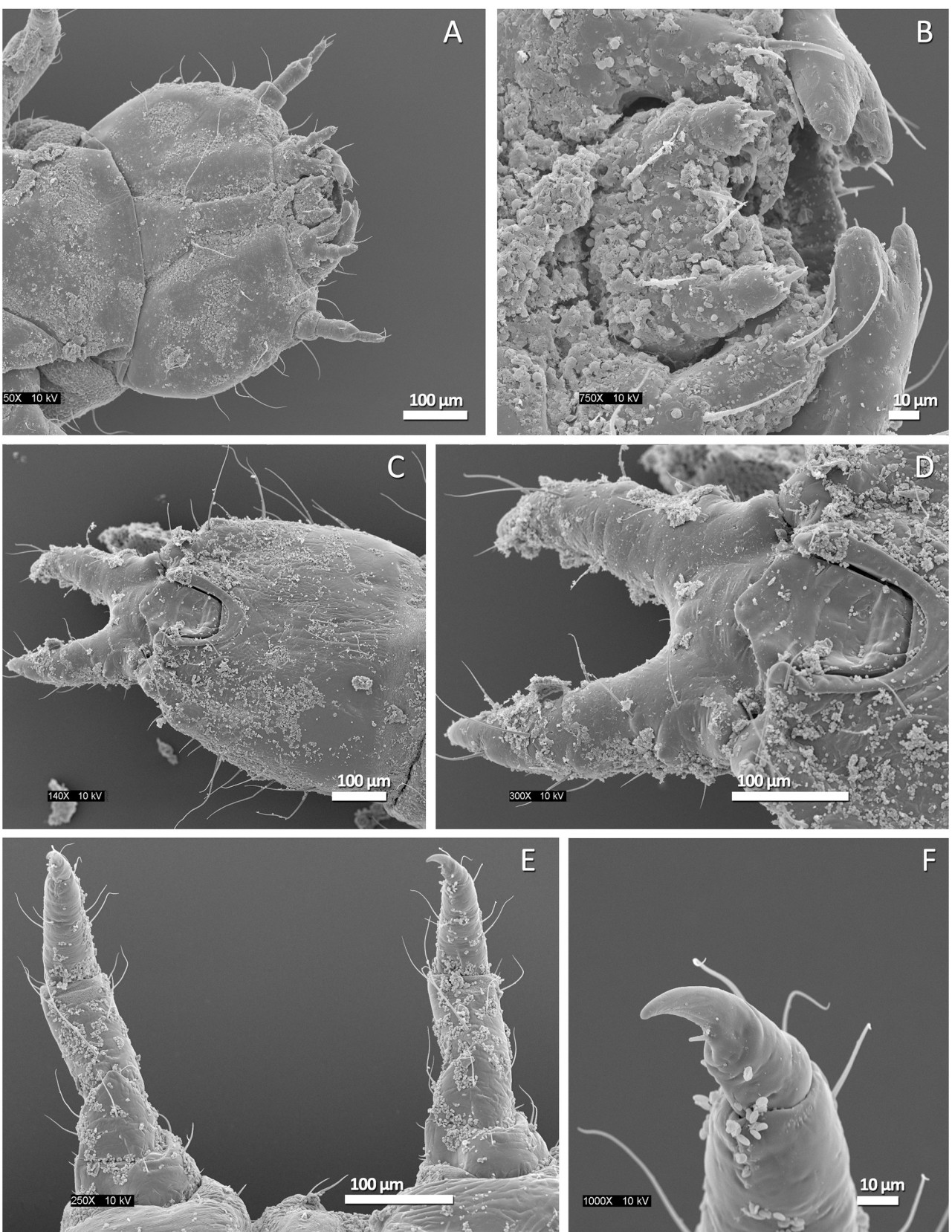

**Fig 7. Representative SEM images of *Laemophloeus biguttatus* body parts in direct contact with the substrate immediately prior to a jump.** A: ventral surface of the head; B: detail of mouthparts; C: Ventral surface of the last abdominal segment and urogomphi; D: detail of last abdominal segment and urogomphi; E: Ventral view of front and middle legs slightly bent inwards; F: Detail of front right tarsal claw. Body surface debris and fungal spores evident in all images.

the first example of such a power amplifying mechanism known to exist in nature. Additional morphological examination to find or rule out the presence of a spring could help to determine which case is true.

This study is one of very few to describe jumping behavior of beetle larvae in the Polyphaga (a group of Coleoptera containing over 340,000 described species; [70]; Table 1) and, to the best of the authors' knowledge, is a unique example of a possible LaMSA or 'LaMMA' mechanism in which the latching component requires interaction with the substrate to function properly. Both the inability of larvae to jump off a smooth surface and the observation that the slip of the leg's grip of the substrate is the first movement in a jump sequence point to the leg-substrate interaction functioning as a latch. Furthermore, it is notable that we were unable to identify any morphological adaptations for latching or jumping from the SEM or microCT data, suggesting that adaptations for jumping in this species may be primarily behavioral (gripping the substrate prior to contracting abdominal muscles, and then releasing grip once enough energy has been elastically stored) rather than morphological, and these beetles may represent an early transitional step between direct muscle actuated movements and a more derived, high performance LaMSA system. This may partially explain the low estimates of power density and long latch-decoupling times for this species compared to other LaMSA systems, including other jumping larvae with highly derived jumping behaviors and morphologies [13, 71–76]. Comparisons with closely related species that have conclusively been shown incapable of jumping to see what morphological characteristics, if any, are derived and may assist in jump performance in this species, as well as identifying and quantifying the characteristics of the spring mechanism, are important next steps in determining whether specific morphological adaptations for jumping are present that we did not detect in our current study.

## Jumping behavior in insect larvae

Larvae that exhibit jumping behavior are found in dozens of species in a variety of ecological contexts (see Table 1 and references within), but there are three distinct circumstances under which the evolution of jumping larvae appears to be favored:

1. **Triungulin/planidial larvae**, i.e. the active, host-seeking first instar of parasitoid species. This includes those strepsipteran, dipteran, and hymenopteran larvae whose first instars appear to use their cercal bristles as a spring to launch themselves onto the host. This seems to be a particularly important strategy for the Acroceridae (Diptera) and Eucharitidae (Hymenoptera), both of which are larval parasites of well-defended predatory arthropods (spiders and ants, respectively). The ability to leap onto a host undetected may be a means of avoiding detection and attack during the larva's dispersal phase.

2. **Encapsulated larvae**, typically insects whose third instar or prepupal stage must seek an appropriate environment for pupation without leaving the seed or leaf envelope in which they have developed. This includes the "Mexican jumping bean" moth *Cydia saltitans*, as well as several other small moth species, sawflies in the genus *Heterarthrus*, cynipid gall wasps, and one species of ichneumonid wasp which parasitizes encapsulated weevil larvae. In this type of legless leaping, the larva braces itself and strikes the inner wall of its gall, seed pod, or leaf envelope hard enough to move the entire capsule. Saeki et al. [38] demonstrated

that *Bathyplectes* larvae are able to direct this seemingly random jumping movement, increasing their activity in sun or heat and coming to rest in shady areas. Similar activity has been documented in other encapsulated species when exposed to light or heat; thus, this type of jumping is very likely a means of moving to a safe pupation habitat without exposing the larva itself to predators.

3. **Larvae at risk of [sudden] exposure**, including those who feed in concealed habitats that are at risk of being disturbed by predators or larger animals. This includes many mycophagous species, fruit and vinegar flies, and fungus gnats. Both the cheese skippers associated with *casu marzu* and their piophilid relatives feeding on vertebrate carcasses display this behavior as well. The common thread among these taxa is that their habitats – fermenting fruit, fungus, and carrion – are ephemeral and also likely to attract other scavengers and predators, particularly vertebrates. Jumping may represent a rapid means of escape from sudden exposure when the food source is disturbed (as described by Brooks and Cotton [42] for larvae of *Conotrachelus anaglypticus*). It was also demonstrated by Bonduriansky [25] that only later stage piophilid larvae jump, in an attempt to move from the food source to suitable pupation sites, thus reducing exposure time. Harvey and Acorn [6] demonstrated that tiger beetle larvae, unearthed from their burrows in loose, sandy soil, react violently to a simulated parasitoid attack by performing what was described by those authors as "leaping somersaults."

## Function of jumping behavior in Laemophloeidae

Due to the cryptic nature of insect larvae living under the bark of decaying plants, their patchy distribution due to ephemeral or sporadic food resources, and few researchers studying their natural history, the behaviors of many subcortical insect larvae are not well known. In fact, while observing the fauna associated with the same tree in which *L. biguttatus* larvae were collected, we collected a number of maggots that also were observed to jump in a species that had not been recorded to do so (pers. obs. by MAB and AAS of *Dasiops vibrissata* Malloch, Lonchaeidae; Table 1).

Although we describe the mechanics of jumping laemophloeid larvae here, one important question remains: why do these larvae jump? It seems very unlikely that the jumping behavior of laemophloeids is used to routinely avoid or repel predators and parasitoids, because of the spatial constraints associated with living under bark or fungal structures. Another piece of evidence against predator/parasite avoidance is the fact that the larvae we observed did not jump when stimulated with forceps or other tools (simulating a predator attack, *cf.* [6]), though they did flail and direct their sharp urogomphi towards the simulated attacker. This was also seen in the *Placonotus* larvae (S1 Video). The larvae instead stopped and jumped after crawling around, without any direct stimulus. The behavior of jumping laemophloeid larvae is most similar to that of mycophagous and saprophagous fly larvae associated with decaying wood and carcasses (Table 1) – a response to sudden exposure, intended to quickly move the insect to a safer microhabitat. Thus we speculate that the function of laemophloeid jumping behavior is to aid in rapid movement to suitable habitats as needed, avoiding predation or parasitism indirectly. We can envision cases where the bark of rotting trees sloughs off easily, exposing the larvae to the elements and attackers. Based on our COT calculations for crawling vs jumping in this species, jumping would result in a more rapid and energetically less costly locomotion compared to crawling (approximately 13% COT for jumping compared to

crawling), and could also produce unpredictable trajectories by which the larvae can escape to new sites.

It is also possible that larval jumping is an artifact or exaptation of another behavior. During our (TY) observations of *Placonotus*, the larvae frequently exhibited a vertical prying action in tight spaces, including subcortical habitats. This behavior appears to facilitate movement under bark or between fungal masses, similar to the "wedge-pushing" of carabid beetles [77], and may use the same musculature as the jumping behavior documented in this study.

Unfortunately, due to the paucity of live specimens for our studies and inability to replicate more natural conditions for them to behave, we cannot fully address this point through experimentation. We encourage future research on this question by collecting larvae of these beetles and performing more experiments.

## Supporting information

**S1 Table. Complete data sets of jump kinematic measures, summarized in Table 2 of the manuscript.**
(XLSX)

**S1 Data. Complete set of tracking coordinates for each analyzed jump and R script used for data analysis.**
(ZIP)

**S1 Video. Real-time (30 frames per second image capture and playback) of jumping behavior observed in *Laemophloeus biguttatus* and *Placonotus testaceus*.** In order of appearance: 1: initial observation of *L. biguttatus* jumping on natural substrate; 2: full *L. biguttatus* jump sequence; 3: additional full *L. biguttatus* jump sequence; 4: closer view of an *L. biguttatus* jump; 5: series of *P. testaceus* jumps off of a tissue paper substrate, filmed from above.
(MP4)

**S2 Video. Slow motion sequences of *Laemophloeus biguttatus* jumping behavior.** In order of appearance: 1: 3,200 frames per second capture of the jump pictured in panels A-C of Fig 3; 2: 3,200 frames per second capture of the jump pictured in panel D of Fig 3; 3: 60,000 frames per second capture of the initiation of jump showing the hind legs detaching from the substrate, as first body movement, when the jump sequence is set into motion.
(MP4)

## Acknowledgments

We thank Alexander Krings (NCSU) for verifying the identification of the oak species, Charles Hodges and Shawn Butler (NCSU Plant Disease and Insect Clinic) for identifying the fungus and amplifying the CO1 gene from the *Laemophloeus* larva, respectively. Valerie K. Lapham and the Center for Electron Microscopy (NCSU) for assistance in SEM imaging. We thank Prof. Andrew Suarez for providing funds to pay for the microCT scan. We thank Prof. Shelia Patek and Justin Jorge for access to and assistance with high-speed camera imaging equipment.

## Author Contributions

**Conceptualization:** Matthew A. Bertone, Joshua C. Gibson, Ainsley E. Seago, Takahiro Yoshida, Adrian A. Smith.

**Formal analysis:** Joshua C. Gibson, Adrian A. Smith.

**Investigation:** Matthew A. Bertone, Joshua C. Gibson, Adrian A. Smith.

**Methodology:** Matthew A. Bertone, Joshua C. Gibson, Adrian A. Smith.

**Writing – original draft:** Matthew A. Bertone, Joshua C. Gibson, Ainsley E. Seago, Takahiro Yoshida, Adrian A. Smith.

**Writing – review & editing:** Matthew A. Bertone, Joshua C. Gibson, Ainsley E. Seago, Takahiro Yoshida, Adrian A. Smith.

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
