## [Decision Letter · Decision Letter 0]

14 Sep 2021

PONE-D-21-25012A Novel Power-Amplified Jumping Behavior in Larval Beetles (Coleoptera: Laemophloeidae)PLOS ONE

Dear Dr. Bertone,

Thank you for submitting your manuscript to PLOS ONE. After careful consideration, we feel that it has merit but does not fully meet PLOS ONE’s publication criteria as it currently stands. Therefore, we invite you to submit a revised version of the manuscript that addresses the points raised during the review process.

There was consensus among all three reviewers that the submission is compelling and should be published. In particular, reviewers were enthusiastic about the integration of multiple data sources in this report of novel locomotion. As you will see in the detailed comments below, each reviewer has offered suggestions to improve the manuscript. Because some of these comments relate to data analysis and interpretation, which may impact results, the determination is "major revision." The most vital suggested revisions involve the interpretation of a putative latch mechanism and inferences made for kinematic calculations. There are also more minor comments related to terminology, methods, and clarity. Please consider all specific comments made by the reviewers carefully and incorporate this feedback into your next submission. If the authors disagree with any particular suggestions or comments, these disagreements should be noted and justified in a response letter.

We look forward to receiving your revised manuscript.

Kind regards,

Phillip Barden

Academic Editor

PLOS ONE

Journal Requirements:

 [TY was partly supported by Research Fellowships of the Japan Society for the Promotion of Science for Young Scientists (JSPS Research Fellowships for Young Scientists, PD: JP19J00167). https://www.jsps.go.jp/english/e-pd/index.html

The funders had no role in study design, data collection and analysis, decision to publish, or preparation of the manuscript.]

[TY was partly supported by Research Fellowships of the Japan Society for the Promotion of Science for Young Scientists (JSPS Research Fellowships for Young Scientists, PD: JP19J00167).]

 [TY was partly supported by Research Fellowships of the Japan Society for the Promotion of Science for Young Scientists (JSPS Research Fellowships for Young Scientists, PD: JP19J00167). https://www.jsps.go.jp/english/e-pd/index.html

The funders had no role in study design, data collection and analysis, decision to publish, or preparation of the manuscript.]

Reviewers' comments:

Reviewer's Responses to Questions

**Comments to the Author**

1. Is the manuscript technically sound, and do the data support the conclusions?

Reviewer #1: Yes

Reviewer #2: Yes

Reviewer #3: Partly

2. Has the statistical analysis been performed appropriately and rigorously? 

Reviewer #1: Yes

Reviewer #2: Yes

Reviewer #3: N/A

3. Have the authors made all data underlying the findings in their manuscript fully available?

Reviewer #1: Yes

Reviewer #2: Yes

Reviewer #3: Yes

4. Is the manuscript presented in an intelligible fashion and written in standard English?

Reviewer #1: Yes

Reviewer #2: Yes

Reviewer #3: Yes

5. Review Comments to the Author

Reviewer #1: This is an excellent paper, documenting a novel mode of animal locomotion, and as such it certainly deserves to be published. I have only a few suggestions (by line number) that I hope the authors will consider before finalizing their text.

Introduction

43: “prelude” jumping behavior—why not “precede,” which is more common usage?

122: Cuticular is presumably a reference to resilin? Why not say resilin? I am not clear on whether other types of cuticle have adequate elastic properties to allow jumping.

Methods

178: What glue was used?

180: Photo information: what aperture was used during videography, and what magnification was achieved (these are more important than maximum aperture and maximum magnification, since they allow the reader to estimate depth of focus). Also, referring to the lens as a Venus Optics product is fine, but the brand name Laowa is more familiar to most people—why not use both?

196: Any idea of how much variance there was in larval mass, even in subjective terms? Mass estimation is important to the calculations that follow.

211: These larvae possess legs, not prolegs, so it is not clear what is being referred to here. Prothoracic legs? Or are all six legs being referred to as prolegs? I suspect that this was an error, since prolegs are fleshy appendages on the abdomen of larval insects.

292: Here, it is not clear how muscle tissue was identified in the MicroCT scans. By location? By shape, or details of structure? By density? Please clarify.

Results

332: I am one of those people who think that “morphology” is the study of structure, and that the word does not refer to structure itself. Thus, the identification was an example of morphology (the study), but it was based on structure. I realize this may seem pedantic.

338: Please provide the compete Genbank identification number.

342: “with reference to the taxonomic literatures” should be rewritten as something such as, “by comparison with published descriptions”

350: Would “pressed” be a better term here than “flattened,” since flattened implies a change in the shape of the head and pygidium? As well, I notice that the head was not at all "flattened" to the substrate, and instead it made contact primarily via the mouthparts, at a fairly steep angle to the substrate.

355: “Circle” is perhaps too precise a description—why not “loop”?

359: I suggest “During” rather than “on”

367: Since no film is used, I suggest sticking with the terms “video capture” or “videographed”

Discussion

418: Why the quotation marks around “leaping somersaults?” I ask not just because it comes from one of my co-authored papers. These behaviours are indeed leaps, since leaps are synonymous with jumps, and since the leaps of tiger beetle larvae involve pitched rotational movement, they are also, literally, somersaults. One might also place “jump” in quotation marks, since most animal jumps involve the hind legs, unlike the jumps you describe. A more obvious metaphor involves the word “spring,” and if anything should be in quotation marks, this is the term. I suggest avoiding quotation marks here, lest they be mistaken for scare quotes, so to speak.

To my eye, one remarkable thing about the laemophloeid larval jumps is the lack of rotational movement. They are remarkably directional in the video clips, and perhaps this should be mentioned.

It also seems possible to me that the rolled-up shape that the larva assumes by the time it returns to the substrate will enable rolling locomotion, as it does with tiger beetle larvae, or at the very least, more or longer bounces.

Figure 6: The basal abdominal segments seem to be important here. There appears to be proportionately less muscle in the basal abdominal segments, or at least the figure gives the impression of a central muscle-poor area on each of these segments. In the initial stage of the jump, the anterior and posterior body regions remain quite straight, while the inflection point occurs just anterior to these segments. Might this help propel the larvae forward, by situating the bend anterior to the center of mass of the larva? These basal abdominal segments are also quite large in these larvae, and one might reasonably imagine that an increase in pressure in the body cavity, followed by expansion of the dorsal regions of the basal abdominal segments, would result in ventral flexion of the body, while the anterior and posterior body regions remain relatively straight because of a combination of pressure and (muscular or cuticular?) rigidity. It seems to me that this paper should propose some such plausible mechanism for power storage and release.

As well, while pondering this review, I was delighted when a preliminary account of this research showed up in my Youtube feed. The video (https://www.youtube.com/watch?v=y-b73G96UIQ) includes additional, excellent slow motion video, and a rotating microCT scan image as well. I found that the video enhanced my understanding of the research, and I suggest that it should be cited in the paper.

Finally, my son Jesse Acorn, who has a background in electrical engineering, wanted me to add that the term “power amplification” promotes a somewhat misleading analogy with electrical amplification, whereas “energy storage” would be a better description of what actually happens in springs and insect bodies.

Reviewer #2: I have only two points to bring up with this manuscript, the first is a small (and somewhat odd) point. The authors do a great job integrating the LaMSA literature as well as discussing the jumping literature; there has just been some recent work on beetle jumping in particular from the Bolmin lab and the Ribak lab (both about adult beetles) which I think could be discussed tying this work to the beetle literature. This is a small suggestion, authors are free to consider or disregard.

My major point is that the authors are using a 400 W/kg threshhold discussion for calling this a power amplified jump; and this threshold is...mushy....This jump behaviour exists in a very odd zone of right near the limit (depending on how much muscle mass is driving the jump; a quantity that is very hard to measure for this animal). I am of the opinion that this jump is worth looking at regardless whether it is power amplified or not (I'm actually fascinated by this particular jump because a latch without a spring could be used to amplify muscle power output just 'as is', just not by a lot - as in, Latch mediated muscle actuated behaviour may be able to generate 400-500 W/kg of energy -depending on parameters in simulations...

The authors do seem aware of this to an extent, just be a bit careful calling this a 'LaMSA' jumper, it may be a 'LaMMA' jumper, which would also be interesting in its own right.

Overall I liked the paper a lot, and just have these minor suggestions for the authors to consider.

Reviewer #3: This manuscript describes the previously undocumented jumping locomotion of beetle larvae, specifically in Laemophloeus biguttatus. This behavior and mechanism of jumping appears to be novel compared to other jumping larvae, since it does not involve forming a loop and creating a latch composed of just the body, instead they hypothesize that this rapid mechanism uses the legs to latch against the substrate (and an unknown spring mechanism) to achieve rapid jumps. The main criteria used to evaluate their hypothesis is the calculation of power density for these jumps compared against a conservative (vertebrate) threshold. In addition, the authors provide a review of jumping in larva to provide some additional insight into the potential reasons why Laemophloeus biguttatus may jump at all.

Overall, I found that this was a strong manuscript, which clearly puts the work into context and does not overstate their findings. I appreciated that the methods (as highlighted in the discussion) were very conservative, both in how power density was calculated (e.g., as a mean power density over the launch phase and with varying amounts of muscle mass included) and with respect to the power density compared against (from a vertebrate known to be selected for high-powered muscular movement). The videos plus their calculations make a convincing case that these movement are probably not entirely muscle actuated, although the details of the LaMSA mechanism are unresolved.

I would like to see more discussion of how they came to their conclusions about the latch mechanism. For instance, what evidence is there from the micro-CT, SEM, or videos that leads the authors to think that the latch is the gripping of the substrate by the larva’s legs and only the legs? Is it possible that there is another interaction (friction or another adhesive mechanism)? It appears from the second supplemental video that there are hairs extending off the ventral surface of the larva on segments without legs. Could these play a role? What is the significance of the fact that larvae could not perform jumps on glass or acrylic?

One concern I had throughout the manuscript was related to the different phases used in the kinematic calculations and descriptions of the jump. Specifically, it appeared that the “latch-decoupling” versus “launch” phases were described in different ways throughout the manuscript. I understand that unlatching can overlap with launch, where unlatching refers to the mediation of elastic potential energy to kinetic energy and launch refers to the period over which a mass is accelerated during a jump. In this manuscript, these phases were initially described based on the movement of the body and placement of the legs or body against the substrate, but sometimes these phases seemed to be described in a different way, were combined for calculations, or were said to overlap to differing degrees.

This confusion made it difficult to evaluate the period over which power was calculated to verify that power (for power density) was calculated over the period during which the jump is being actuated (when mass is being accelerated). For instance, it appears that power was calculated over the combined duration of the latch-decoupling and launch phases- so this would mean that it was calculated over the light blue period in shown in Figure 5 (C? There are no letters on the figure). However, from this panel, it appears that acceleration of the center of mass has already begun before the blue period, so part of the launch has not been included in the calculation, based on my interpretation of this plot. In addition, from this plot, it would appear that the latch-decoupling phases is entirely part of the launch phase, since the mass is accelerating.

For an example of conflicting definitions of these phases, I refer to the paragraph beginning with line 394. In the first line, the launch phase is defined to start when all legs have released (line 394), yet this paragraph goes on to say that sometimes the legs are the last body part in contact with the substrate (Line 400). How did this affect what was included in the launch phase? Was time with a leg but no other body parts against the substrate included in the mean power calculated for this jump? I bring this up because ultimately differences in the duration of time included in the calculation of mean power will have a substantial impact on the estimated power densities, since P is proportional to t^(-3). I have ultimately suggested “major revision” because changes made to the definition of the actuation phase could require recalculating power and other kinematics reported in the manuscript.

Also note that unlatching phases, though rarely reported, are generally extremely short in duration, even in comparison to spring actuation/launching phases. Part of this is because the efficient transfer of potential to kinetic energy should require a very rapid unlatching mechanism. Unlatching more slowly will dissipate more energy (Divi et al. 2020). This is one reason that the comparatively long latching-decoupling phase as described here seems odd as an unlatching phase.

Finally, I would recommend that the authors make sure that these phases are clearly defined and used consistently throughout the manuscript, performing any new analyses that are necessary. The launch phase is critical to the calculation of power and could be more clearly defined kinematically as the time over which the center of mass is accelerating (probably from onset of rapid bending to the moment that all parts of the larva leave the substrate). I would recommend deemphasizing, potentially removing the “latch-decoupling phase” and/or simply renaming this phase as something other than a latching phase, until there is further evidence that leg-substrate contact is as the latch. Leg contact is still interesting and, if anything, I would appreciate a better understanding of leg contact relative to other parts of the jump across the strikes recorded.

Below I have listed more specific points:

Methods: What were the average sizes of each species of larva? How far behind the larva was the grid placed that was used to calibrate the kinematics? Based on the size of the animal, they have a fairly course resolution because of 0.5cm grid calibration, and the fact that it was not a calibration placed in the exact plane that the animal was in could affect measured displacement and thus power.

Line 237: Can you explain your reasoning for assigning displacement to your pre-airborne phases more clearly and make it clear which phases are overlapping?

Figure 3: How do loading, release, and launch phases as pictured relate to phases as described in the methods? In particular when does latching and unlatching occur? Does release refer to spring release, latch release, or just release of the legs? What is inferred to be occurring during this phase and how do you know it doesn’t belong as part of the loading phase or, alternatively, as part of the launch phase? Note that launch as defined in Longo et al. 2019 and Farley et al. 2019 should be the spring actuated (powered) phase and should not include time when the animal is fully airborne since there should no longer be a way to increase energy in the system once contact with the substrate is lost. As pictured in C, launch appears to include time in the air. Note that the language used to describe the sequence is different in Figure 5: e.g. “release phase” is described as “latch-decoupling phase.” Adopting a uniform set of terms throughout the manuscript would help the reader understand these different kinematic phases.

Line 271- 273: The value reported from Askew and Marsh is the highest average power density reported for a vertebrate, but not the highest power density reported (for instance instantaneous measurements of power density are much higher, >1000 W/kg muscle). From the methods and equation (5) it looks the average this is the correct benchmark to use for comparison, since it appears that power has been calculated over a duration of time. Throughout the manuscript, please refer to these power densities as the maximum average power density to make this distinction clear.

Line 86-87: Needs citation “Subsequent studies have found that larval jumping is widespread in holometabolous insects, including Lepidoptera, Hymenoptera, and Diptera.”

Larval identification: The methods and results for this should be combined. It was distracting not to know which species came from which location until the results section.

Line 61: How was this comparison done (e.g., BLAST search?). What published sequence did your match to and how strong was the match?

Line 197: Probably a typo with an unneeded “as” in “Average mass was used for all jump calculations as due to the sensitivity…”

Table 1: How were jump pitch, roll, and yaw calculated? I did not see this described in the methods. 3D kinematics should require another view (camera or mirror) of the jump.

Figure 3: A-C should have a scale bar. Could times be placed next to each larval image to better understand how each is related to the others? A horizontal timeline could also be another alternative here, especially because the overlap in phases could be shown more clearly.

Citation for articles referred to and not cited in the manuscript:

Divi, S., Ma, X., Ilton, M., St. Pierre, R., Eslami, B., Patek, S. N., & Bergbreiter, S. (2020). Latch-based control of energy output in spring actuated systems. Journal of the Royal Society Interface, 17(168), 20200070.

6. PLOS authors have the option to publish the peer review history of their article (what does this mean?). If published, this will include your full peer review and any attached files.

Reviewer #1: **Yes: **John H. Acorn

Reviewer #2: No

Reviewer #3: No

---

## [Author Response · Author response to Decision Letter 0]

26 Nov 2021

PONE-D-21-25012

A Novel Power-Amplified Jumping Behavior in Larval Beetles (Coleoptera: Laemophloeidae)

PLOS ONE

Dear editor(s).

Please see our responses to the reviewer comments below.

 [TY was partly supported by Research Fellowships of the Japan Society for the Promotion of Science for Young Scientists (JSPS Research Fellowships for Young Scientists, PD: JP19J00167). https://www.jsps.go.jp/english/e-pd/index.html

The funders had no role in study design, data collection and analysis, decision to publish, or preparation of the manuscript.]

[TY was partly supported by Research Fellowships of the Japan Society for the Promotion of Science for Young Scientists (JSPS Research Fellowships for Young Scientists, PD: JP19J00167).]

- Thank you. We have removed this statement from the manuscript.

 [TY was partly supported by Research Fellowships of the Japan Society for the Promotion of Science for Young Scientists (JSPS Research Fellowships for Young Scientists, PD: JP19J00167). https://www.jsps.go.jp/english/e-pd/index.html

The funders had no role in study design, data collection and analysis, decision to publish, or preparation of the manuscript.]

- We have made all of our data available through additional supplementary files

Reviewer #1: This is an excellent paper, documenting a novel mode of animal locomotion, and as such it certainly deserves to be published. I have only a few suggestions (by line number) that I hope the authors will consider before finalizing their text.

Introduction

43: “prelude” jumping behavior—why not “precede,” which is more common usage?

- We have changed it to “precede”

122: Cuticular is presumably a reference to resilin? Why not say resilin? I am not clear on whether other types of cuticle have adequate elastic properties to allow jumping.

- We have changed to mention resilin and resilin composite materials, with added citations

Methods

178: What glue was used?

- Specific brand info added to the methods

180: Photo information: what aperture was used during videography, and what magnification was achieved (these are more important than maximum aperture and maximum magnification, since they allow the reader to estimate depth of focus). Also, referring to the lens as a Venus Optics product is fine, but the brand name Laowa is more familiar to most people—why not use both?

- We have added all of the above suggestions and details to the methods description

196: Any idea of how much variance there was in larval mass, even in subjective terms? Mass estimation is important to the calculations that follow.

- We did not have access to a fine enough scale to report individual body masses. However, we do report the range of body size/lengths observed. Our kinematic power estimates are very conservative so any small changes due to a range of masses would not change the conclusions we reach in the manuscript.

211: These larvae possess legs, not prolegs, so it is not clear what is being referred to here. Prothoracic legs? Or are all six legs being referred to as prolegs? I suspect that this was an error, since prolegs are fleshy appendages on the abdomen of larval insects.

- Correct, that was a typo, we have edited to “leg”

292: Here, it is not clear how muscle tissue was identified in the MicroCT scans. By location? By shape, or details of structure? By density? Please clarify.

- We clarify that muscles were identified based on their general shape and location within the body as well as increased contrast compared to other morphological features as a result of I2E staining.

Results

332: I am one of those people who think that “morphology” is the study of structure, and that the word does not refer to structure itself. Thus, the identification was an example of morphology (the study), but it was based on structure. I realize this may seem pedantic.

- It’s fine. We edited the statement, using “morphologically” to describe the identification and “anatomy” as an alternative to “structure” as suggested

338: Please provide the compete Genbank identification number.

- The Genbank numbers have been provided

342: “with reference to the taxonomic literatures” should be rewritten as something such as, “by comparison with published descriptions”

- We have updated as described.

350: Would “pressed” be a better term here than “flattened,” since flattened implies a change in the shape of the head and pygidium? As well, I notice that the head was not at all "flattened" to the substrate, and instead it made contact primarily via the mouthparts, at a fairly steep angle to the substrate.

- Thanks for the clarification. We have edited the wording to match your observations.

355: “Circle” is perhaps too precise a description—why not “loop”?

- “circle” has been changed to “loop”

359: I suggest “During” rather than “on”

- Edited as suggested

367: Since no film is used, I suggest sticking with the terms “video capture” or “videographed”

- Edited based on suggestion

Discussion

418: Why the quotation marks around “leaping somersaults?” I ask not just because it comes from one of my co-authored papers. These behaviours are indeed leaps, since leaps are synonymous with jumps, and since the leaps of tiger beetle larvae involve pitched rotational movement, they are also, literally, somersaults. One might also place “jump” in quotation marks, since most animal jumps involve the hind legs, unlike the jumps you describe. A more obvious metaphor involves the word “spring,” and if anything should be in quotation marks, this is the term. I suggest avoiding quotation marks here, lest they be mistaken for scare quotes, so to speak.

- Edited so as not to be confused for “scare quotes”, but retaining that the authors’ described the behavior in this way.

To my eye, one remarkable thing about the laemophloeid larval jumps is the lack of rotational movement. They are remarkably directional in the video clips, and perhaps this should be mentioned.

- We added a line to note this in the results and referred readers to table 1 where we reported rotational movement data.

It also seems possible to me that the rolled-up shape that the larva assumes by the time it returns to the substrate will enable rolling locomotion, as it does with tiger beetle larvae, or at the very least, more or longer bounces.

- We agree. However, the platforms we used to film the behavior were narrow (to keep the beetles in the focal plane) and post-jump rolling often led them off the platform. So, we are unable to analyze that aspect of this behavioral sequence. We have added a line to the results to include consideration of post-jump rolling and to mention that we did not analyze that aspect.

Figure 6: The basal abdominal segments seem to be important here. There appears to be proportionately less muscle in the basal abdominal segments, or at least the figure gives the impression of a central muscle-poor area on each of these segments. In the initial stage of the jump, the anterior and posterior body regions remain quite straight, while the inflection point occurs just anterior to these segments. Might this help propel the larvae forward, by situating the bend anterior to the center of mass of the larva? These basal abdominal segments are also quite large in these larvae, and one might reasonably imagine that an increase in pressure in the body cavity, followed by expansion of the dorsal regions of the basal abdominal segments, would result in ventral flexion of the body, while the anterior and posterior body regions remain relatively straight because of a combination of pressure and (muscular or cuticular?) rigidity. It seems to me that this paper should propose some such plausible mechanism for power storage and release.

- It is the authors’ opinion that discerning the exact mechanism of the jump is beyond the scope of our current manuscript and warrants additional future studies. In addition, the functional significance of muscle distribution between segments is difficult to discern without data on the morphology of non-jumping relatives for comparison. Future work that examines the anatomy of multiple jumping and non-jumping species, as well as a more detailed study of the loading phase of jumps could be done to uncover potential loading mechanisms.

As well, while pondering this review, I was delighted when a preliminary account of this research showed up in my Youtube feed. The video (https://www.youtube.com/watch?v=y-b73G96UIQ) includes additional, excellent slow motion video, and a rotating microCT scan image as well. I found that the video enhanced my understanding of the research, and I suggest that it should be cited in the paper.

- We are delighted that the reviewer saw and appreciated the YouTube video of this study! However, we do not wish to cite the video in the paper as the permanence of YouTube URLs is not something under our control. Furthermore, the narrative format of the research story on YouTube is written and edited for a primarily public, non-peer scientist audience. We feel that the format of this research story here, as we present it in this paper, is the proper one to have archived in the scientific record.

Finally, my son Jesse Acorn, who has a background in electrical engineering, wanted me to add that the term “power amplification” promotes a somewhat misleading analogy with electrical amplification, whereas “energy storage” would be a better description of what actually happens in springs and insect bodies.

- The authors acknowledge that power amplification has discrepant usage between biologists and engineers, and the field is generally moving towards the LaMSA framework as one way to rectify this. However, since power amplification has an established definition in biology that is relevant to the analyses we performed, we believe it is still worth including this terminology in our manuscript. If the reviewer is insistent that we change this, we can oblige.

Reviewer #2: I have only two points to bring up with this manuscript, the first is a small (and somewhat odd) point. The authors do a great job integrating the LaMSA literature as well as discussing the jumping literature; there has just been some recent work on beetle jumping in particular from the Bolmin lab and the Ribak lab (both about adult beetles) which I think could be discussed tying this work to the beetle literature. This is a small suggestion, authors are free to consider or disregard.

- We appreciate the information and suggestions, but we think it’s best to keep the focus of this paper on larval “jumping” and not include information about adult jumping behaviors (larval and adult beetles being very different).

My major point is that the authors are using a 400 W/kg threshhold discussion for calling this a power amplified jump; and this threshold is...mushy....This jump behaviour exists in a very odd zone of right near the limit (depending on how much muscle mass is driving the jump; a quantity that is very hard to measure for this animal). I am of the opinion that this jump is worth looking at regardless whether it is power amplified or not (I'm actually fascinated by this particular jump because a latch without a spring could be used to amplify muscle power output just 'as is', just not by a lot - as in, Latch mediated muscle actuated behaviour may be able to generate 400-500 W/kg of energy -depending on parameters in simulations...

The authors do seem aware of this to an extent, just be a bit careful calling this a 'LaMSA' jumper, it may be a 'LaMMA' jumper, which would also be interesting in its own right.

- We have added a discussion of the ‘LaMMA’ possibility to the discussion section of the manuscript. 

Overall I liked the paper a lot, and just have these minor suggestions for the authors to consider.

Reviewer #3: This manuscript describes the previously undocumented jumping locomotion of beetle larvae, specifically in Laemophloeus biguttatus. This behavior and mechanism of jumping appears to be novel compared to other jumping larvae, since it does not involve forming a loop and creating a latch composed of just the body, instead they hypothesize that this rapid mechanism uses the legs to latch against the substrate (and an unknown spring mechanism) to achieve rapid jumps. The main criteria used to evaluate their hypothesis is the calculation of power density for these jumps compared against a conservative (vertebrate) threshold. In addition, the authors provide a review of jumping in larva to provide some additional insight into the potential reasons why Laemophloeus biguttatus may jump at all.

Overall, I found that this was a strong manuscript, which clearly puts the work into context and does not overstate their findings. I appreciated that the methods (as highlighted in the discussion) were very conservative, both in how power density was calculated (e.g., as a mean power density over the launch phase and with varying amounts of muscle mass included) and with respect to the power density compared against (from a vertebrate known to be selected for high-powered muscular movement). The videos plus their calculations make a convincing case that these movement are probably not entirely muscle actuated, although the details of the LaMSA mechanism are unresolved.

I would like to see more discussion of how they came to their conclusions about the latch mechanism. For instance, what evidence is there from the micro-CT, SEM, or videos that leads the authors to think that the latch is the gripping of the substrate by the larva’s legs and only the legs? Is it possible that there is another interaction (friction or another adhesive mechanism)? It appears from the second supplemental video that there are hairs extending off the ventral surface of the larva on segments without legs. Could these play a role? What is the significance of the fact that larvae could not perform jumps on glass or acrylic?

- We have added statements in the results and discussion regarding the inability of larvae to perform jumps on glass and acrylic. MAB also looked at numerous other specimens of beetle larvae that live in the same type of habitat, and with similar morphology, and found setae like the ones found on Laemophloeus on multiple non-jumping species. Furthermore, the simple and delicate structure (i.e. not thickened or spatulate) of these setae do not correspond to a specialized function that might be used for jumping. We do not think these setae are significant so we have not added anything in the text concerning them. 

One concern I had throughout the manuscript was related to the different phases used in the kinematic calculations and descriptions of the jump. Specifically, it appeared that the “latch-decoupling” versus “launch” phases were described in different ways throughout the manuscript. I understand that unlatching can overlap with launch, where unlatching refers to the mediation of elastic potential energy to kinetic energy and launch refers to the period over which a mass is accelerated during a jump. In this manuscript, these phases were initially described based on the movement of the body and placement of the legs or body against the substrate, but sometimes these phases seemed to be described in a different way, were combined for calculations, or were said to overlap to differing degrees.

- We have attempted to clarify this throughout the manuscript.

This confusion made it difficult to evaluate the period over which power was calculated to verify that power (for power density) was calculated over the period during which the jump is being actuated (when mass is being accelerated). For instance, it appears that power was calculated over the combined duration of the latch-decoupling and launch phases- so this would mean that it was calculated over the light blue period in shown in Figure 5 (C? There are no letters on the figure). However, from this panel, it appears that acceleration of the center of mass has already begun before the blue period, so part of the launch has not been included in the calculation, based on my interpretation of this plot. In addition, from this plot, it would appear that the latch-decoupling phases is entirely part of the launch phase, since the mass is accelerating.

- The authors have carefully re-checked the frames of each video and confirmed that the light blue + purple (the latch-decoupling and launch phases) correspond to the period when the beetles are accelerating based on a rapid change in how quickly the beetle’s body moves upward in the videos starting at this time. The slight discrepancy in the acceleration plot is likely due to smoothing errors in the displacement spline (likely caused by slight tracking error due to the beetle changing orientation over the course of the jump, which would slightly change the centroid’s position during auto-tracking) magnified by taking the second derivative to generate the acceleration plot. Decreasing the smoothing factor narrows the acceleration peak so that it falls within the latch decoupling + launch phases, but results in an overall more jagged acceleration plot. We do not believe this would affect the power or power density calculations, just how the graph looks visually. The reviewer is correct that the latch-decoupling phase is entirely part of the launch phase in this jump, we have made changes throughout the manuscript to make the distinction between how we defined these phases, how they correspond to the actual acceleration period of the jump, and how we combined phases for calculations of power and power density to correct for this. 

For an example of conflicting definitions of these phases, I refer to the paragraph beginning with line 394. In the first line, the launch phase is defined to start when all legs have released (line 394), yet this paragraph goes on to say that sometimes the legs are the last body part in contact with the substrate (Line 400). How did this affect what was included in the launch phase? Was time with a leg but no other body parts against the substrate included in the mean power calculated for this jump? I bring this up because ultimately differences in the duration of time included in the calculation of mean power will have a substantial impact on the estimated power densities, since P is proportional to t^(-3). I have ultimately suggested “major revision” because changes made to the definition of the actuation phase could require recalculating power and other kinematics reported in the manuscript.

- We have edited this paragraph to improve clarity. As mentioned earlier, latch-decoupling and launch times and displacement were always combined for these calculations as there was always acceleration occurring while some but not all of the legs had lost contact with the substrate. The initial description of the latch-decoupling and launch phases were based on an idealized jump where the legs all release their grip on the substrate near instantaneously, but this rarely occurred in actual jumps, and occasionally all but one leg would release and the beetle would initially accelerate but then ‘stall” until the last leg finally let go, resulting in weaker jumps than if all legs released their grip at once.

Also note that unlatching phases, though rarely reported, are generally extremely short in duration, even in comparison to spring actuation/launching phases. Part of this is because the efficient transfer of potential to kinetic energy should require a very rapid unlatching mechanism. Unlatching more slowly will dissipate more energy (Divi et al. 2020). This is one reason that the comparatively long latching-decoupling phase as described here seems odd as an unlatching phase.

- We agree that the lengthy latch decoupling phase is a bit unusual. To address this we have expanded our discussion of the evidence for PA/LaMSA to discuss how these longer times both help explain the low power density estimations observed compared to other LaMSA systems and may be the result of the beetles being an early step in the evolution of more derived LaMSA systems, in which a latch mechanism is present but is imperfect and results in substantial energy loss while still amplifying power to an extent. 

Finally, I would recommend that the authors make sure that these phases are clearly defined and used consistently throughout the manuscript, performing any new analyses that are necessary. The launch phase is critical to the calculation of power and could be more clearly defined kinematically as the time over which the center of mass is accelerating (probably from onset of rapid bending to the moment that all parts of the larva leave the substrate). I would recommend deemphasizing, potentially removing the “latch-decoupling phase” and/or simply renaming this phase as something other than a latching phase, until there is further evidence that leg-substrate contact is as the latch. Leg contact is still interesting and, if anything, I would appreciate a better understanding of leg contact relative to other parts of the jump across the strikes recorded.

- We attempted to clarify the definition of these phases and their usage throughout the manuscript. 

Below I have listed more specific points:

Methods: What were the average sizes of each species of larva? 

- This was/is reported in column 2 of table 2

How far behind the larva was the grid placed that was used to calibrate the kinematics? Based on the size of the animal, they have a fairly course resolution because of 0.5cm grid calibration, and the fact that it was not a calibration placed in the exact plane that the animal was in could affect measured displacement and thus power.

- We report the width of the platform on which we placed the beetle (2cm) in the methods; this was affixed to the backing board with the 0.5cm grid. Thus the maximum distance the animal could initiate a jump from our scale was 2cm. We understand the reviewers points here, and ideally, yes, the jumps would be all perfectly perpendicular to the camera and the grid and scale would exactly match the plane on which the behavior occurred. However, we used unrestrained animals and tried to record jumps as naturally as possible. That meant giving the animal space in which to move freely and keeping the backboard scale far enough away so that the animal would not jump into it on most of its jumps, yet close enough to get a fairly accurate scale. Matching camera angle and scale with the exact plane of the jump was therefore impossible. 

Line 237: Can you explain your reasoning for assigning displacement to your pre-airborne phases more clearly and make it clear which phases are overlapping?

- We have added a more detailed explanation of how and why the displacement occurring during the latch decoupling and launch phases were pooled, and how this might further bias our estimates of power and power density.

Figure 3: How do loading, release, and launch phases as pictured relate to phases as described in the methods? In particular when does latching and unlatching occur? Does release refer to spring release, latch release, or just release of the legs? What is inferred to be occurring during this phase and how do you know it doesn’t belong as part of the loading phase or, alternatively, as part of the launch phase? Note that launch as defined in Longo et al. 2019 and Farley et al. 2019 should be the spring actuated (powered) phase and should not include time when the animal is fully airborne since there should no longer be a way to increase energy in the system once contact with the substrate is lost. As pictured in C, launch appears to include time in the air. Note that the language used to describe the sequence is different in Figure 5: e.g. “release phase” is described as “latch-decoupling phase.” Adopting a uniform set of terms throughout the manuscript would help the reader understand these different kinematic phases.

- We thank the reviewer for pointing out some confusing aspects of this figure caused by an unclear figure legend. The topmost image of panel C is not included as part of the launch phase; the end of the launch phase was always recorded as the last frame where any part of the larvas’ bodies were still in contact with the ground. We have also updated the phase terminology used in the figure caption to correspond with the uniform terminology used throughout the rest of the manuscript.

Line 271- 273: The value reported from Askew and Marsh is the highest average power density reported for a vertebrate, but not the highest power density reported (for instance instantaneous measurements of power density are much higher, >1000 W/kg muscle). From the methods and equation (5) it looks the average this is the correct benchmark to use for comparison, since it appears that power has been calculated over a duration of time. Throughout the manuscript, please refer to these power densities as the maximum average power density to make this distinction clear.

- We have clarified that this value corresponds to the maximum average power density rather than the maximum instantaneous power density throughout the manuscript, where it is mentioned.

Line 86-87: Needs citation “Subsequent studies have found that larval jumping is widespread in holometabolous insects, including Lepidoptera, Hymenoptera, and Diptera.”

- We now cite the table summarizing jumping larvae with references within.

Larval identification: The methods and results for this should be combined. It was distracting not to know which species came from which location until the results section.

- We think that there are equal arguments for keeping these separate as there are with combining these sections, so we prefer to keep the manuscript how it is. Methods for identification and the results of an identification are different things.

Line 61: How was this comparison done (e.g., BLAST search?). What published sequence did your match to and how strong was the match?

- Information about this is in the results section. 

Line 197: Probably a typo with an unneeded “as” in “Average mass was used for all jump calculations as due to the sensitivity…”

- Edited to correct grammar 

Table 1: How were jump pitch, roll, and yaw calculated? I did not see this described in the methods. 3D kinematics should require another view (camera or mirror) of the jump.

- We added a line to the methods to explain that this was manually recorded for each jump. An additional view was not needed to estimate in-air body rotations.

Figure 3: A-C should have a scale bar. Could times be placed next to each larval image to better understand how each is related to the others? A horizontal timeline could also be another alternative here, especially because the overlap in phases could be shown more clearly.

- The sequence used for panels A-C was filmed for illustrative purposes only, therefore no scale was included in that filming set and no accurate size scale is available for that sequence. We have added timecode labels to the individual beetle images in that sequence to be clearer about the timeline of sequential images we are showing. Figure legend has been updated to reflect this change.

Citation for articles referred to and not cited in the manuscript:

Divi, S., Ma, X., Ilton, M., St. Pierre, R., Eslami, B., Patek, S. N., & Bergbreiter, S. (2020). Latch-based control of energy output in spring actuated systems. Journal of the Royal Society Interface, 17(168), 20200070.

- We have included a citation of this paper in our added discussion of the effects of latch-decoupling duration on performance in addressing other comments above.

---

## [Editor Report · Decision Letter 1]

7 Dec 2021

A novel power-amplified jumping behavior in larval beetles (Coleoptera: Laemophloeidae)

PONE-D-21-25012R1

Dear Dr. Bertone,

We’re pleased to inform you that your manuscript has been judged scientifically suitable for publication and will be formally accepted for publication once it meets all outstanding technical requirements.

Kind regards,

Phillip Barden

Academic Editor

PLOS ONE
---

## [Editor Report · Acceptance letter]

13 Dec 2021

PONE-D-21-25012R1 

A novel power-amplified jumping behavior in larval beetles (Coleoptera: Laemophloeidae) 

Dear Dr. Bertone:

I'm pleased to inform you that your manuscript has been deemed suitable for publication in PLOS ONE. Congratulations! Your manuscript is now with our production department. 

Kind regards, 

on behalf of

Dr. Phillip Barden 

Academic Editor

PLOS ONE